# Transcranial direct current stimulation modulates primate brain dynamics across states of consciousness

**Guylaine Hoffner**[1]\*‡, **Pablo Castro**[1,2]‡, **Lynn Uhrig**[1,3], **Camilo Miguel Signorelli**[4,5,6], **Morgan Dupont**[1], **Jordy Tasserie**[7], **Alain Destexhe**[2], **Rodrigo Cofre**[2,8], **Jacobo Sitt**[9]†, **Béchir Jarraya**[1,10]\*†

[1]Cognitive Neuroimaging Unit, CEA, INSERM, Université Paris-Saclay, NeuroSpin Center, Gif sur Yvette, France; [2]Institute of Neuroscience (NeuroPSI), Paris-Saclay University, Centre National de la Recherche Scientifique (CNRS), Gif sur Yvette, France; [3]Department of Anesthesiology and Critical Care, Necker Hospital, AP-HP, Université de Paris Cité, Paris, France; [4]Department of Computer Science, University of Oxford, Oxford, United Kingdom; [5]Laboratory of Neurophysiology and Movement Biomechanics (LNMB), Université Libre de Bruxelles (ULB), Brussels, Belgium; [6]Center for Philosophy of Artificial Intelligence, University of Copenhagen, Copenhagen, Denmark; [7]Center for Brain Circuit Therapeutics, Department of Neurology, Brigham & Women's Hospital, Harvard Medical School, Boston, United States; [8]Centre Inria Université Côte d'Azur, Equipe Cronos, France; [9]Sorbonne Université, Institut du Cerveau - Paris Brain Institute - ICM, Inserm, CNRS, Paris, France; [10]Department of Neurology, Hopital Foch, Suresnes, France

**\*For correspondence:** guylaine.hoffner@inserm.fr (GH); bechir.jarraya@gmail.com (BJ)

†These authors contributed equally to this work

‡These authors also contributed equally to this work

**Competing interest:** The authors declare that no competing interests exist.

**Sent for Review** 07 August 2024

**Preprint posted** 08 August 2024

**Reviewed preprint posted** 04 October 2024

**Reviewed preprint revised** 27 August 2025

**Version of Record published** 13 October 2025

## eLife Assessment

This **valuable** study applies transcranial direct current stimulation (tCDS) to the prefrontal cortex of non-human primates during two states: (1) propofol-induced unconsciousness; and (2) wakeful performance of a fixation task. The analysis offers **incomplete** evidence to indicate that the effect of tDCS on brain dynamics, as recorded with functional magnetic resonance imaging, is contingent on the state of consciousness during which the stimulation is applied. The findings will be of interest to researchers interested in brain stimulation and consciousness.

**Abstract** The resting primate brain is traversed by spontaneous functional connectivity patterns that show striking differences between conscious and unconscious states. Transcranial direct current stimulation (tDCS), a non-invasive neuromodulatory technique, can improve signs of consciousness in disorders of consciousness (DOCs); however, can it influence both conscious and unconscious dynamic functional connectivity? We investigated the modulatory effect of prefrontal cortex (PFC) tDCS on brain dynamics in awake and anesthetized non-human primates using functional MRI. In awake macaques receiving either anodal or cathodal tDCS, we found that cathodal stimulation robustly disrupted the repertoire of functional connectivity patterns, increased structure–function correlation (SFC), decreased Shannon entropy, and favored transitions toward anatomically based patterns. Under deep sedation, anodal tDCS significantly altered brain pattern distribution and reduced SFC. The prefrontal stimulation also modified dynamic connectivity arrangements typically associated with consciousness and unconsciousness. Our findings offer compelling evidence that PFC tDCS induces striking modifications in the fMRI-based dynamic organization of the brain across different states of consciousness. This study contributes to an

enhanced understanding of tDCS neuromodulation mechanisms and has important clinical implications for DOCs.

## Introduction

At rest, the spontaneous activity of the awake brain is highly structured. The brain dynamically transitions through various resting-state functional magnetic resonance imaging (rs-fMRI) patterns, which are thought to support specific cognitive functions (*Deco et al., 2013*; *Fox et al., 2006*; *Vincent et al., 2006*; *Damoiseaux et al., 2006*; *Raichle et al., 2001*; *Calhoun et al., 2014*; *Preti et al., 2017*; *Baker et al., 2014*; *Deco et al., 2011*). Interestingly, in non-human primates (NHPs), the induction of wakefulness and unconsciousness by different anesthetics is associated with strikingly different sequences and occurrences of transient brain patterns (*Barttfeld et al., 2015*; *Uhrig et al., 2018*). The conscious brain is traversed by a diverse and mobile set of functional brain patterns that deviate from anatomical connectivity. In contrast, the unconscious brain is characterized by transient functional patterns that are less diverse, stiffer, and more closely follow anatomical connectivity. Similar dynamic functional configurations are found in awake and sedated humans and generalize to other contexts of loss of consciousness, whether in patients with impaired consciousness or during sleep (*Luppi et al., 2021*; *Huang et al., 2020*; *Demertzi et al., 2019*; *Golkowski et al., 2019*; *Castro et al., 2024b*). Ultimately, these different structure–function configurations are proposed as signatures of consciousness and unconsciousness, respectively.

Over the past 20 years, transcranial direct current stimulation (tDCS) has emerged as a popular non-invasive tool for clinicians and researchers to modulate brain activity (*Narmashiri and Akbari, 2025*; *Lefaucheur et al., 2017*). Two or more electrodes are attached to the subject's head, and a weak current is applied between them, producing an electric field, part of which crosses the skull and influences the activity of underlying neurons (*Lefaucheur and Wendling, 2019*). In vivo and in vitro data suggest that tDCS acts through diverse, concurrent, and multiscale modes of action (*Lefaucheur et al., 2017*; *Lefaucheur and Wendling, 2019*). Importantly, ongoing neural activity during tDCS administration is likely to be critical, as tDCS effects are state-dependent and primarily modulate active networks (*Antal et al., 2007*; *Bikson et al., 2013*; *Li et al., 2019*).

Recent studies suggest that tDCS may influence the level of consciousness. Anodal tDCS of the left dorsolateral prefrontal cortex (DLPFC) improves signs of consciousness in patients with disorders of consciousness (DOCs) (*Thibaut et al., 2014*; *Angelakis et al., 2014*; *Hermann et al., 2020*; *Liu et al., 2023*) and can increase rs-fMRI connectivity in some of these patients, both at the site of stimulation and in remote regions (*Peng et al., 2022*). However, the effects of prefrontal cortex (PFC) tDCS on resting-state dynamic brain patterns which are typically associated with consciousness have never been investigated. In healthy humans, tDCS can perturb both occurrences and transitions of some rs-fMRI co-activation patterns (CAPs) (using the DLPFC as a seed), suggesting that some elements of brain dynamics could be affected by tDCS in the awake resting brain. Anesthesia provides a unique model for studying consciousness, which, similarly to DOC, is characterized by the disruption or even the loss of consciousness (*Luppi, 2024*). Additionally, anesthesia mechanisms involve several subcortical nuclei that are key components of the brain's sleep and arousal circuits (*Kelz and Mashour, 2019*). The effects of tDCS on brain activity during anesthesia-induced loss of consciousness remain largely unexplored. Repeated stimulation of the rat motor cortex facilitates the return of the righting reflex and visual and working memory after isoflurane anesthesia (*Mansouri and García, 2021*), suggesting that tDCS may represent a novel approach to accelerate recovery from general anesthesia (*Kato and Solt, 2021*).

However, these studies left many questions unanswered. Does PFC tDCS possess the ability to remodel the functional and structural brain organization characteristic of the conscious and unconscious brain? For example, can it affect the order and relative frequency of fMRI brain connectivity patterns, as well as other aspects of brain dynamics, such as the structure–function correlation (SFC) and the Shannon entropy (*Demertzi et al., 2019*)? If so, do the polarity (e.g., anodal vs. cathodal), the intensity (e.g., 1 vs. 2 mA), and the timing of the stimulation (e.g., during vs. after the stimulation) perturb these metrics differently? Ultimately, does tDCS have enough perturbational power to affect the dynamic configurations of connectivity characteristic of consciousness and unconsciousness

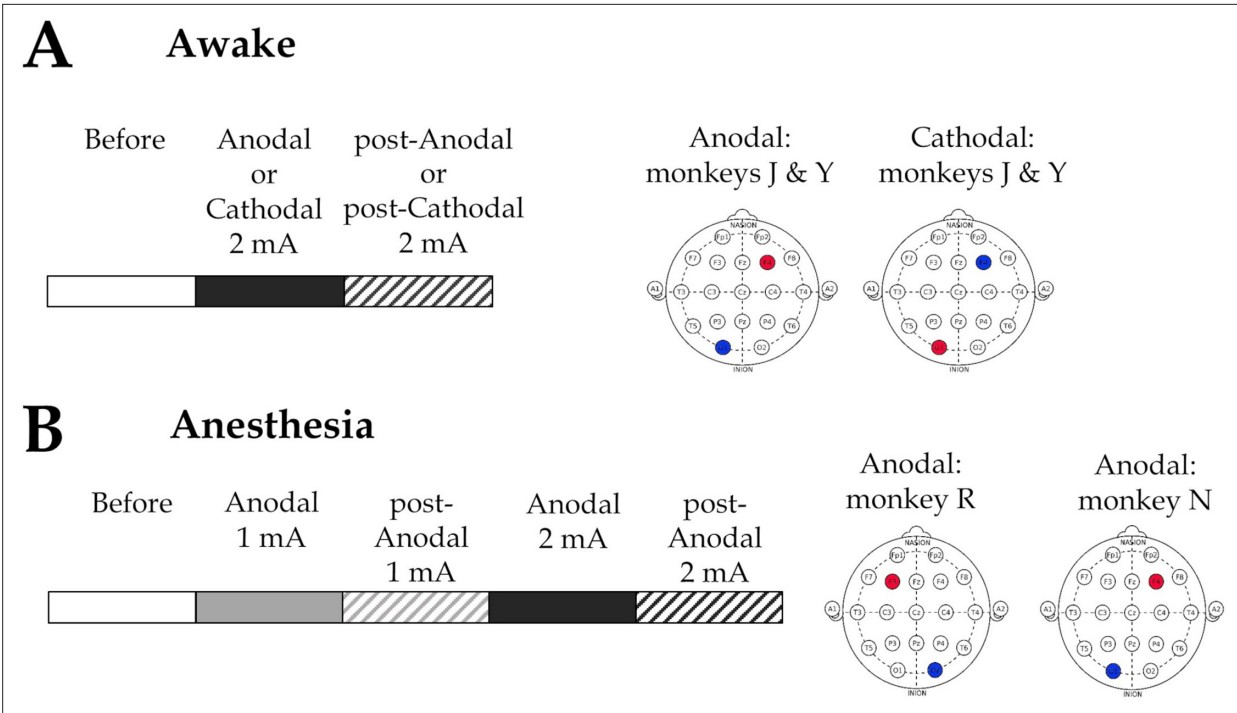

**Figure 1.** Schematic representation of the experimental designs and transcranial direct current stimulation (tDCS) electrode montages employed in the study. (**A**) Awake experiments. fMRI data were acquired before, during, and after 20 min of tDCS stimulation at 2 mA intensity. The anodal electrode (red) was either placed over the right prefrontal cortex (PFC; F4) and the cathodal electrode (blue) over the left occipital cortex (O1) ('anodal' montage), or a 'reversed' montage was used, with the cathodal electrode over F4 and the anodal electrode over O1 ('cathodal' montage). (**B**) Anesthesia experiments. fMRI data were recorded before, during, and after 20 min of consecutive 1 and 2 mA tDCS stimulation. An anodal electrode montage was employed, targeting the left or right PFC (F4/O1 or F3/O2) depending on the animal and its anatomical constraints. Before: before stimulation. Anodal and post-Anodal: during and after anodal tDCS of the PFC. Cathodal and post-Cathodal: during and after cathodal tDCS of the PFC.

The online version of this article includes the following figure supplement(s) for figure 1:

**Figure supplement 1.** Structural images displaying electrode placements on the head of monkeys.

residing in cortical brain dynamics? Answering these questions may both advance our comprehension of the neural mechanisms underlying tDCS effects and provide meaningful information for clinicians.

This study aimed to determine whether tDCS targeting the PFC can modify the relationship between the structural and functional brain organization in fMRI signals in NHPs, either in the awake state or under deep propofol anesthesia. Our specific aims were to determine the effects of PFC tDCS on brain dynamics (1) in awake NHPs receiving either anodal or cathodal stimulation, (2) in deeply anesthetized NHPs receiving anodal stimulation at two different intensities, and (3) the time course of these putative changes relative to the stimulation (i.e., during and after the stimulation). Our results provide evidence that anodal prefrontal tDCS had only a marginal effect on brain dynamics in awake animals. In contrast, cathodal prefrontal tDCS significantly reorganized the functional repertoire in favor of anatomically similar brain patterns without apparent behavioral modification. Moreover, both anodal and cathodal stimulation affected the Markov chain transitions between brain patterns, with some changes specific to ongoing stimulation and others to the post-stimulation period. Under anesthesia, anodal prefrontal tDCS delivered at an intensity of 2 mA, but not 1 mA, significantly reduced the SFC but did not fully restore the broad and rich dynamic repertoire of functional patterns characteristic of wakefulness, nor did it induce signs of arousal in the animals. Altogether, these findings illustrate the ability of tDCS to modulate conscious and unconscious brain dynamics in a polarity- and intensity-dependent manner.

**Table 1.** Number of fMRI runs acquired in each condition for each monkey, in awake and deeply sedated states.

| Awake state | Monkey J | Monkey Y |
|---|---|---|
| Before tDCS | 76 | 60 |
| Anodal 2 mA tDCS | 32 | 24 |
| Post-Anodal 2 mA tDCS | 32 | 24 |
| Cathodal 2 mA tDCS | 12 | 8 |
| Post-Cathodal 2 mA tDCS | 12 | 8 |
| **Anesthetized state** | **Monkey R** | **Monkey N** |
| Before tDCS | 32 | 8 |
| Anodal 1 mA tDCS | 16 | 8 |
| Post-Anodal 1 mA tDCS | 16 | 8 |
| Anodal 2 mA tDCS | 16 | 8 |
| Post-Anodal 2 mA tDCS | 16 | 8 |

## Results

We established an experimental procedure that allowed tDCS and fMRI data acquisition from macaque monkeys (*Macaca mulatta*) in two states of consciousness: awake and under general anesthesia (*Figure 1* and *Table 1*). We also monitored eye fixation rates in the awake state (*Table 2*) and vital parameters in the anesthesia state (*Table 3*).To characterize the functional and structural brain organization in fMRI signals and how they are affected by the stimulation, we calculated the dynamic functional connectivity between 82 cortical regions of interest (ROIs) based on the CoCoMac anatomical database. We used the Hilbert transform to estimate the phase of fMRI signals at each repetition time (TR) and computed the coordination between the continuous fMRI time series across all ROIs. We then applied the unsupervised $k$-means algorithm to cluster the phase-based coherence matrices. This allowed for the identification of recurrent patterns of brain connectivity. Finally, we sorted the brain patterns based on their degree of similarity to the CoCoMac anatomical matrix. We calculated the relative occurrence rates of the patterns, the SFC, and the Shannon entropy for each vigilance state, electrode arrangement, and stimulation condition. A graphical overview of the overall analysis is shown in *Figure 2*.

In *Figures 3–6*, we present the results for brain patterns (or $k$ clusters) fixed to 6, as this number was determined to be the best choice for maximizing the inter-pattern correlation variance (IPCV) when considering awake and anesthesia datasets together (*Figure 2—figure supplement 1*). Additionally, we verified that varying the number of brain patterns (or $k$ clusters) in the $k$-means algorithm

**Table 2.** Percentage of eye fixation during transcranial direct current stimulation (tDCS) fMRI experiments in the awake state.

| Condition | Animal | Fixation (%) |
|---|---|---|
| Before tDCS | Monkey J | 99 ± 2 |
| | Monkey Y | 98 ± 4 |
| Anodal 2 mA tDCS | Monkey J | 99 ± 3 |
| | Monkey Y | 99 ± 2 |
| Post-Anodal 2 mA tDCS | Monkey J | 98 ± 3 |
| | Monkey Y | 98 ± 3 |
| Cathodal 2 mA tDCS | Monkey J | 99 ± 1 |
| | Monkey Y | 100 ± 1 |
| Post-Cathodal 2 mA tDCS | Monkey J | 100 ± 1 |
| | Monkey Y | 100 ± 1 |

**Table 3.** Physiological data during transcranial direct current stimulation (tDCS) fMRI experiments under deep propofol anesthesia.

Physiological parameters under deep propofol anesthesia and different tDCS stimulation conditions: heart rate; oxygen saturation ($SpO_2$); systolic blood pressure (SAP); diastolic blood pressure (DAP); respiration rate; end-tidal $CO_2$.

| Condition | Heart rate | $SpO_2$ | SAP | DAP | Respiration rate | EtCO$_2$ |
|---|---|---|---|---|---|---|
| Before tDCS | 127 (10) | 97 (3) | 116 (11) | 60 (7) | 20 (2) | 40 (1) |
| Anodal 1 mA tDCS | 123 (8) | 98 (1) | 118 (9) | 60 (4) | 19 (2) | 40 (1) |
| Post-Anodal 1 mA tDCS | 120 (7)* | 98 (1) | 118 (9) | 61 (4) | 19 (2) | 39 (1) |
| Anodal 2 mA tDCS | 120 (7)* | 98 (1) | 120 (6) | 63 (3) | 19 (2) | 39 (1) |
| Post-Anodal 2 mA tDCS | 123 (4) | 98 (1) | 123 (4)* | 65 (3)* | 19 (2) | 40 (1) |

A one-way ANOVA revealed that there was a statistically significant difference in mean heart rate between at least two conditions ($F(4, 111) = [3.64]$, p = [0.008]). The Bonferroni test for multiple comparisons found that the mean value of heart rate in the post-stimulation 1 mA and stimulation 2 mA conditions was significantly lower compared to pre-stimulation ((p = 0.024, 95% C.I. = [0.52,13.05]) and (p = 0.021, 95% C.I. = [0.62,13.15]), respectively). There was also a statistically significant difference in mean systolic and diastolic blood pressure between at least two conditions (($F(4, 131) = [2.97]$, p = [0.022]) and ($F(4, 130) = [3.07]$, p = 0.018), respectively). The Bonferroni test showed that the mean systolic and diastolic blood pressures for the 2 mA post-stimulation condition were significantly higher compared to the pre-stimulation condition ((p = 0.017, 95% C.I. = [−13.51,−0.77]) and (p = 0.033, 95% C.I. = [−8.36,−0.20]), respectively).

*Statistically significant difference compared to pre-stimulation, at p < 0.05 with Bonferroni correction. In brackets, the standard deviation.

from 3 to 10 did not change the principal findings in the paper (*Figure 2—figure supplements 2–4*). Boxplots displaying the main measurables (slope's coefficient and Shannon entropy) as a function of said *k*-number of clusters were also done (figure supplement 1 of *Figures 3 and 5–7*) to further demonstrate the consistency of our results, regardless of the choice of *k*.

## Polarity-dependent effects of prefrontal tDCS on the cortical dynamic connectivity in awake macaques

To examine whether tDCS can perturb the repertoire of functional connectivity patterns characteristic of the awake state, we collected a total of 288 fMRI runs before, during, and after the application of tDCS at an intensity of 2 mA in two macaque monkeys using two different electrode setups (*Figure 1A*; *Table 1*). Representative structural images showing electrode placements on the head of the two awake monkeys are shown in *Figure 1—figure supplement 1A*. *Supplementary file 1* displays the complete set of structural images, showing that the two electrodes were accurately placed over the PFC and the occipital cortex in a reproducible manner across awake sessions.

We performed the dynamic connectivity analysis on data from all five awake experimental conditions: before tDCS, during tDCS using anodal or cathodal montages, and after anodal or cathodal tDCS (*Figure 3*). The resulting patterns exhibited a variety of complex configurations of interregional connectivity. These included long-range positive/negative coherences (patterns 1 and 3), mostly frontal positive and parieto-occipital negative coherences (patterns 2 and 4), and almost strictly positive medium or low interregional coherences (patterns 5 and 6) (*Figure 3A*). We then calculated the probability distribution of the occurrence of each pattern across the stimulation conditions (*Figure 3C*). Brain pattern 6, referred to as the 'anatomical pattern' due to its highest correlation with the CoCoMac connectome, showed the most numerous and strongest differences in visit frequency. The time spent in this pattern increased sharply during and following stimulation with the cathodal montage compared to before stimulation and to the corresponding conditions with the anodal montage. This later finding shows the specificity of the effects of the cathodal stimulation. The time spent in brain state 5 was markedly shorter in the cathodal post-stimulation condition compared to

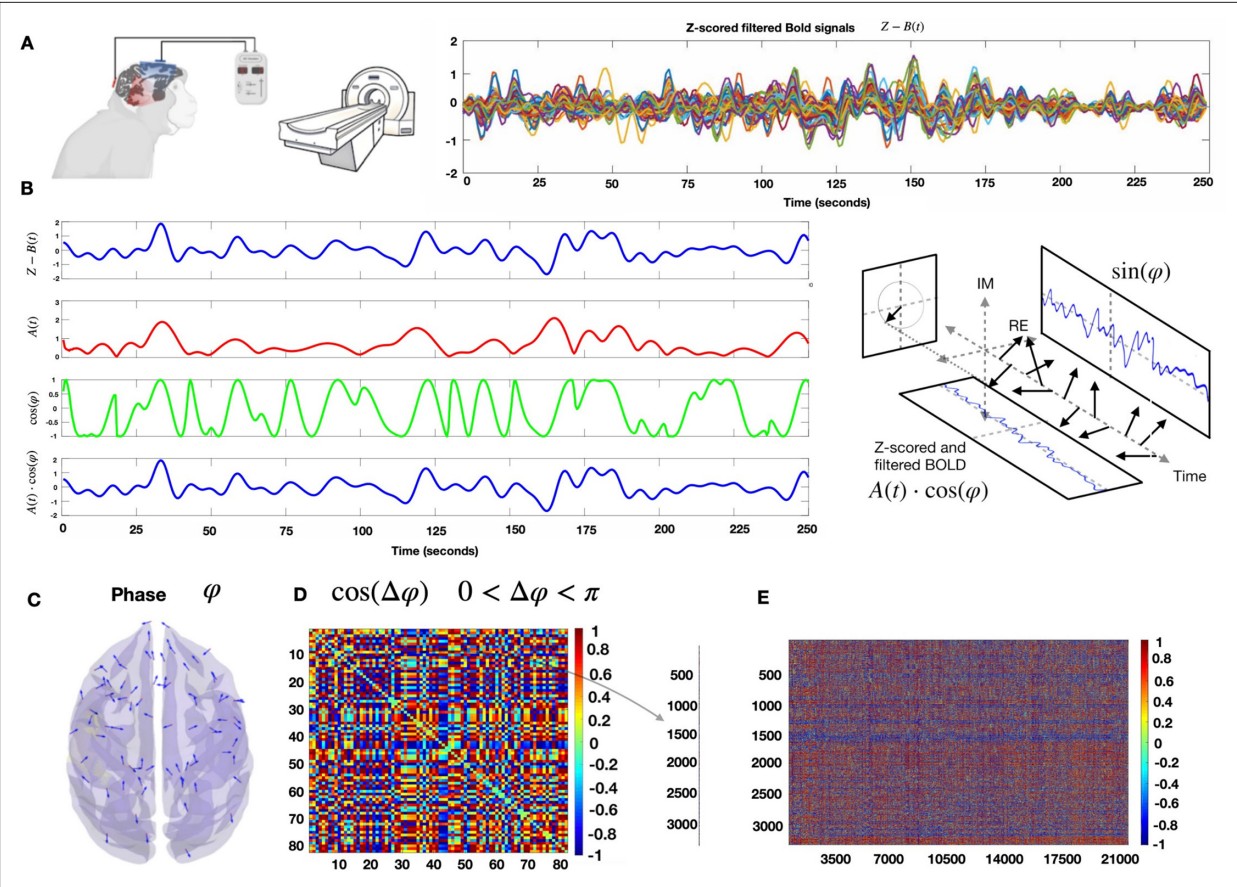

**Figure 2.** fMRI phase coherence analysis. (**A**) Left: Animals were scanned before, during, and after prefrontal cortex (PFC) transcranial direct current stimulation (tDCS) in the awake state (two macaques) or under deep propofol anesthesia (two macaques). Right: Example of z-scored filtered BOLD time series for one macaque. (**B**) Hilbert transform of the z-scored BOLD signal of one region of interest (ROI) into its time-varying amplitude $A(t)$ (red) and the real part of the phase $\varphi$ (green). In blue, we recover the original z-scored BOLD signal as $A(t)\cos(\varphi)$. (**C**) Example of the phase of the Hilbert transform for each brain region at one TR. (**D**) Symmetric matrix of cosines of the phase differences between all pairs of brain regions. (**E**) We extract the superior triangular half of the phase coherence matrix and vectorize it. All time points of the recordings from the different conditions are concatenated together.

The online version of this article includes the following figure supplement(s) for figure 2:

**Figure supplement 1.** Inter-pattern correlation variance (IPCV), a measure used to identify the optimal number of *k* clusters (or brain patterns).

**Figure supplement 2.** Dynamical functional patterns from awake and anesthesia datasets.

**Figure supplement 3.** Dynamical functional patterns from the awake dataset.

**Figure supplement 4.** Dynamical functional patterns from the anesthesia dataset.

pre-stimulation. Cathodal stimulation also shortened the time spent in brain state 3 during stimulation compared to the corresponding condition with the anodal montage. In sharp contrast, the distribution of pattern occupancies during or following stimulation with the anodal montage did not exhibit much change compared to pre-stimulation. These polarity-specific effects of prefrontal tDCS were further illustrated by plotting the occurrence rate per condition (*Figure 3D*). These results suggest that the cathodal stimulation lowered the occurrence of patterns with high frontal coherence (patterns 1, 3, and 5), but promoted the visit of patterns with predominant parieto-occipital coherence (patterns 2 and 4) or very low coherence (pattern 6). It should be noted that these alterations of brain dynamics were not accompanied by any obvious behavioral changes in the monkeys, as they continued to show consistent and optimal eye fixation rates (*Table 2*). All significant statistics are shown in *Table 4*.

Next, we computed the Shannon entropy from the normalized histograms of occurrences (*Figure 3E*, *Table 4*). The Shannon entropy was considerably lower during and after cathodal stimulation compared to before stimulation and during cathodal stimulation compared to during anodal

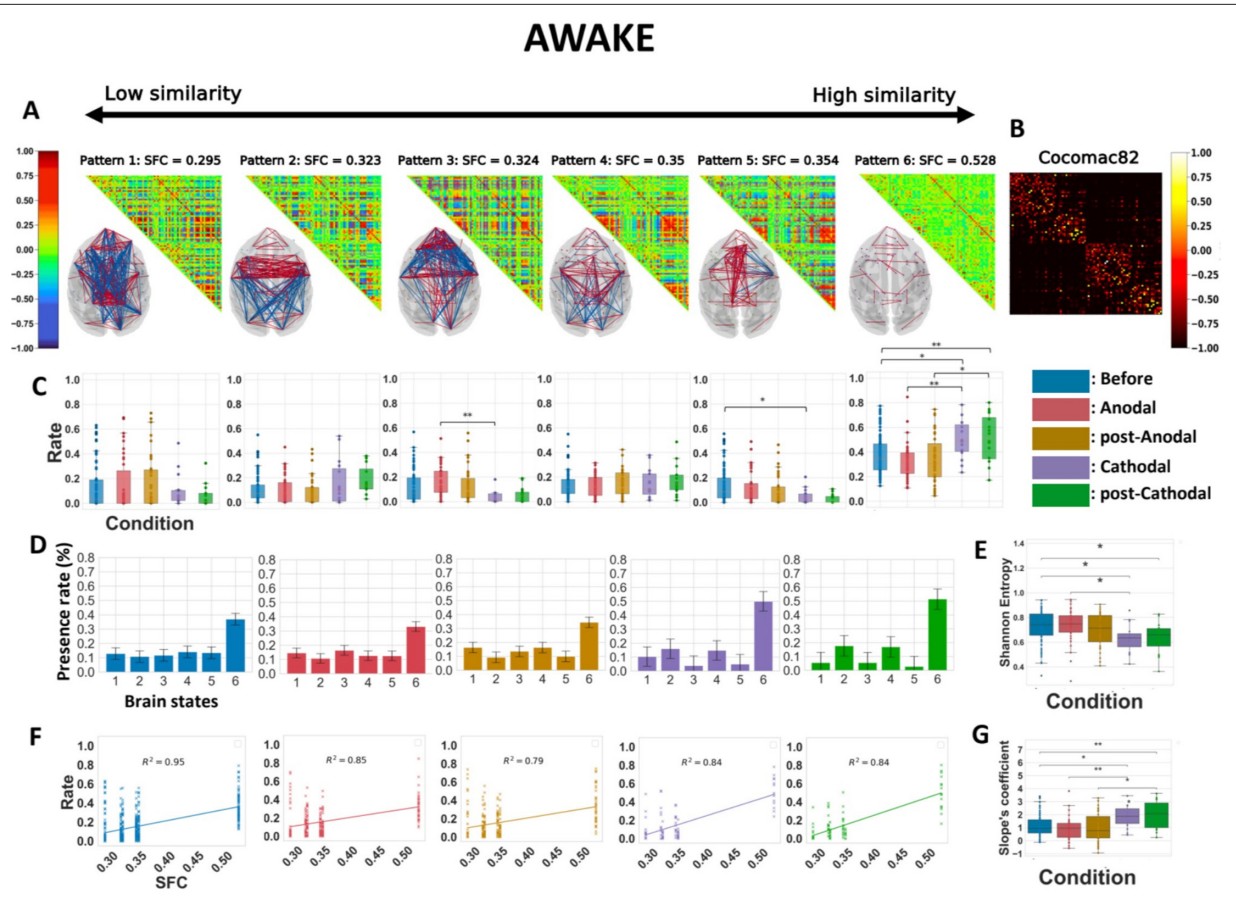

**Figure 3.** Cathodal but not anodal transcranial direct current stimulation (tDCS) of the prefrontal cortex (PFC) alters the repertoire of functional brain patterns, increases structure–function coupling, and decreases Shannon entropy in awake animals. (**A**) Matrix representation of the brain patterns for $k = 6$, obtained by including all the five awake conditions. Brain coordination patterns are ranked from least (left) to highest (right) correlation to the structural connectivity matrix. Networks are plotted in anatomical space (transverse view), showing only values $<-0.5$ and $>0.5$. Red lines represent positive synchronizations between regions of interest, and blue lines represent negative ones. (**B**) Macaque connectome in the CoCoMac82 parcellation. (**C**) Rate of occurrence of each brain connectivity pattern across the five conditions. Boxplots show median occurrence rates with interquartile range and maximum–minimum values (whiskers). (**D**) Barplots of the six brain states and their average presence rate over the different conditions. (**E**) Normalized Shannon entropy as a function of the conditions. (**F**) Rates of occurrence of brain patterns as a function of their similarity in functional and structural connectivity (SFC) for the five conditions. Lines are calculated, per conditions, based on the best linear fit between the average presence rate of each pattern and their SFC. $R^2$, Spearman correlation. Crosses represent individuals' presence rates. (**G**) Coefficient of the linearly regressed slope across conditions. Before: pre-stimulation. Anodal and post-Anodal: during and after anodal tDCS of the PFC (F4/O1 montage). Cathodal and post-Cathodal: during and after cathodal tDCS of the PFC (O1/F4 montage). Asterisks indicate statistically significant differences between conditions (*: $p < 0.05$; **: $p < 0.01$).

The online version of this article includes the following figure supplement(s) for figure 3:

**Figure supplement 1.** Slope and Shannon entropy in the awake conditions.

**Figure supplement 2.** Dynamical functional connectivity analysis of the awake dataset with $k = 4$ numbers of clusters. (**A-G**) as in Figure 3.

**Figure supplement 3.** Variance in inter-region phase coherences of brain patterns.

stimulation. In contrast, the Shannon entropy was not affected by stimulation with the anodal montage, further supporting the polarity-dependent nature of the effects. This result implies that cathodal, but not anodal stimulation limits the degree of surprise held in the distribution of pattern rates. A lower Shannon entropy was reported in many different studies investigating anesthesia-induced loss of consciousness in humans (*Golkowski et al., 2019*; *Castro et al., 2024b*), in macaques (*Barttfeld et al., 2015*; *Uhrig et al., 2018*), under sleep-induced loss of consciousness in humans (*Patel et al., 2020*; *Castro et al., 2024b*), and in patients suffering from DOCs (*Demertzi et al., 2019*). Moreover, recovery from general anesthesia in humans (*Patel et al., 2020*; *Castro et al., 2024b*) and restoration

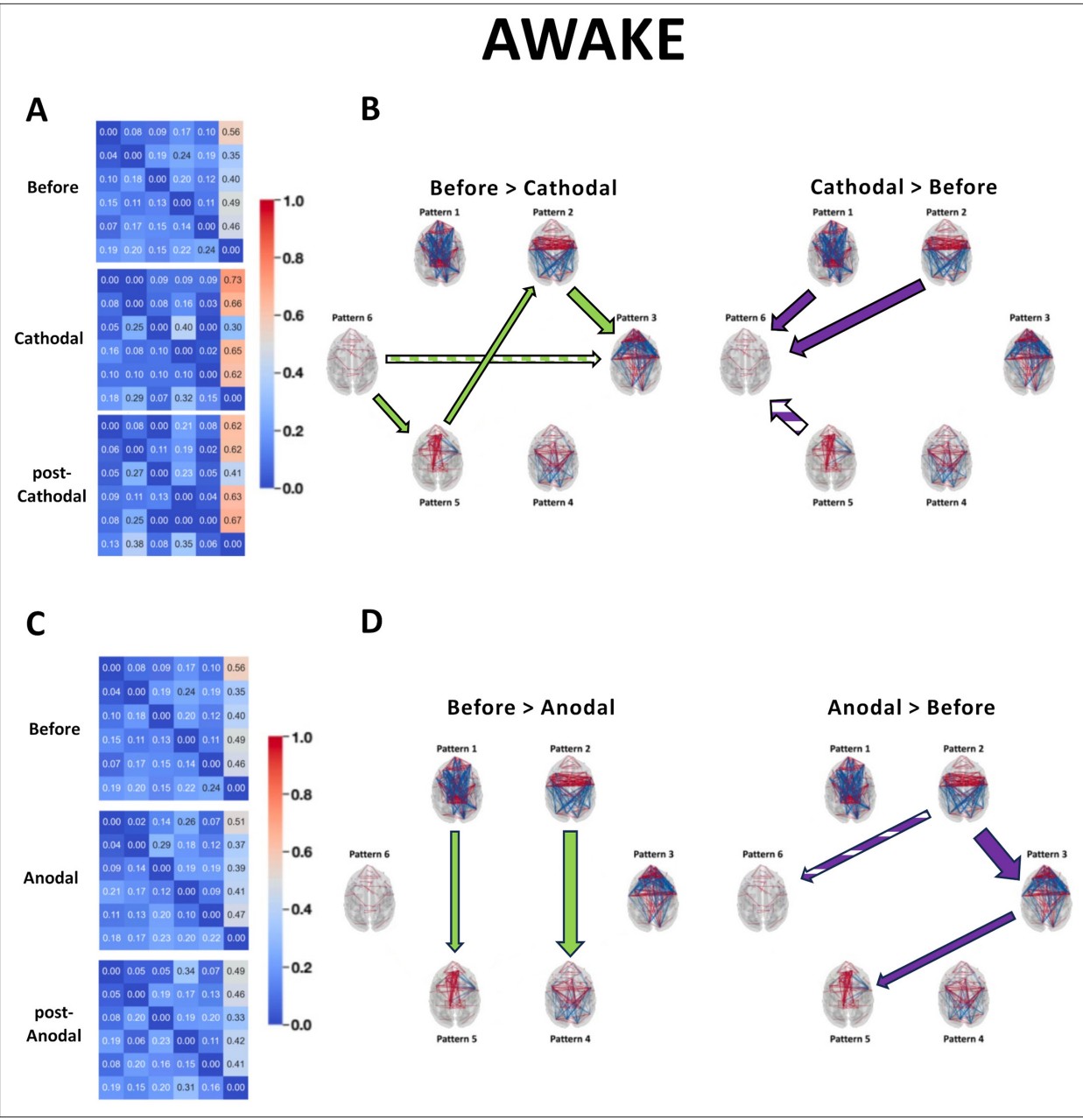

**Figure 4.** Transition probabilities and Markov chain analysis of the identified brain states during and after cathodal or anodal prefrontal cortex (PFC) stimulation in the awake state. Overall transition probabilities matrices pre-stimulation, during, and after cathodal stimulation (**A**) or during and after anodal stimulation (**C**). (**B**, **D**) Left panels: transition probabilities greater before stimulation than during stimulation (full green arrows) or post-stimulation (hashed green arrows). Right panels: transition probabilities greater during (full purple arrows) or after (hashed purple arrows) the stimulation than before. Arrow size indicates p-value significance. Before: pre-stimulation. Cathodal and post-Cathodal: during and after cathodal transcranial direct current stimulation (tDCS) of the PFC (O1/F4 montage). Anodal and post-Anodal: during and after anodal tDCS of the PFC (F4/O1 montage).

of arousal and wakefulness in anesthetized macaques by deep brain stimulation of the central thalamus (*Tasserie et al., 2022*; *Luppi et al., 2024*) were both associated with recovery of a higher Shannon entropy value.

We then evaluated the relationship between the centroids' presence rate and the anatomical backbone in monkeys. We plotted the presence rates of all the centroids, for all macaques, as a function of their SFC. Considering the averages of occurrence rates over the different conditions, we computed one linear regression (per condition), to show the overall linear relationship between the time spent in the various brain states and their similarity to the anatomical connectivity for each condition

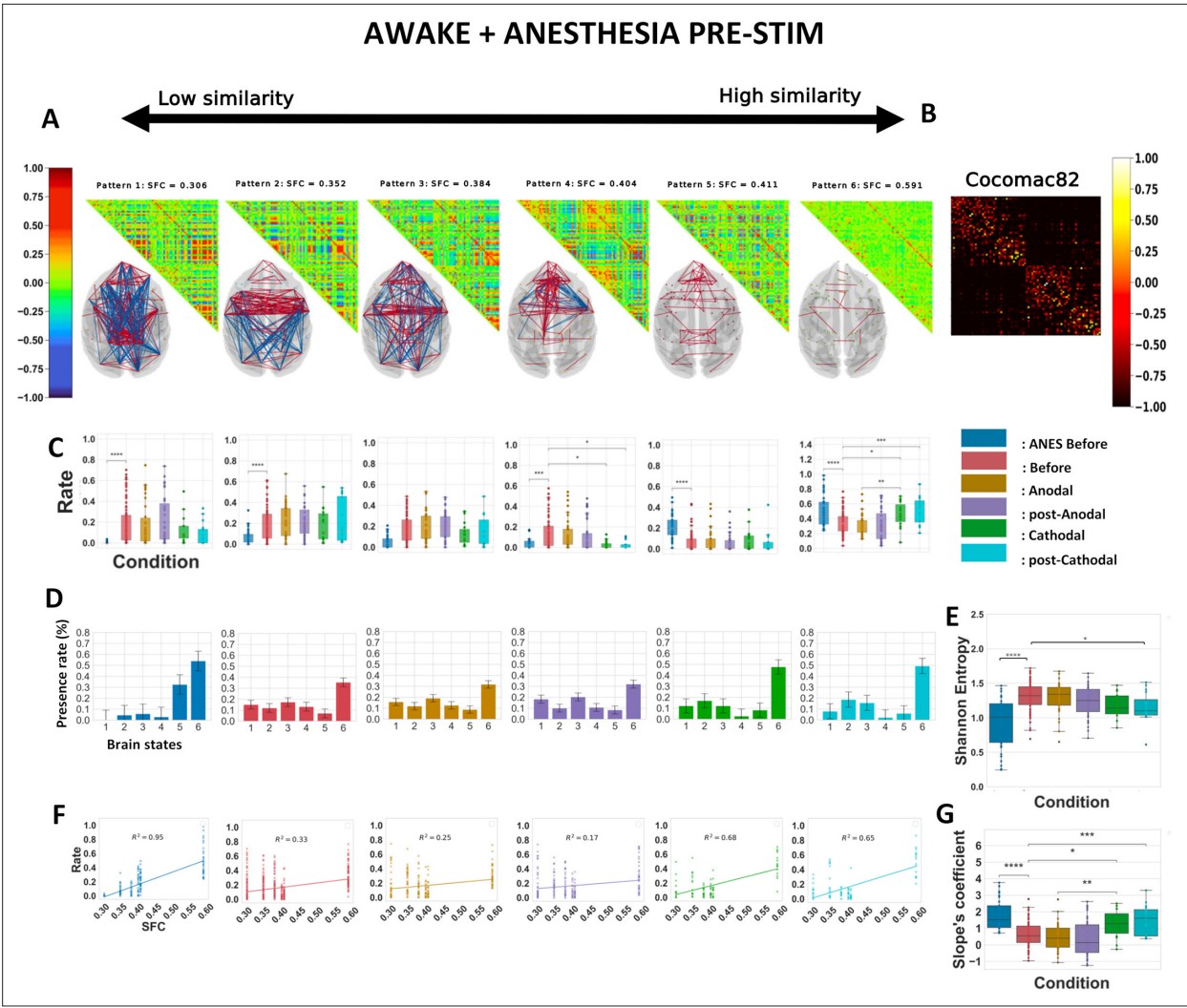

**Figure 5.** The polarity-dependent effects of prefrontal transcranial direct current stimulation (tDCS) on awake brain dynamics resist contrasting with anesthetized dynamics. Similar analyses to those shown in *Figure 3* (panels **A–G**) are presented here for $k = 6$, considering all five awake conditions as well as the anesthesia baseline condition. ANES before: before stimulation under anesthesia. Before: pre-stimulation. Anodal and post-Anodal: during and after anodal tDCS of the PFC (F4/O1 montage). Cathodal and post-Cathodal: during and after cathodal tDCS of the PFC (O1/F4 montage). Similar analysis. Asterisks indicate statistically significant differences between conditions (*: $p < 0.05$; **: $p < 0.01$; ***: $p < 0.001$; ****: $p < 0.0001$).

The online version of this article includes the following figure supplement(s) for figure 5:

**Figure supplement 1.** Slope and Shannon entropy in awake conditions contrasted with anesthesia pre-stimulation condition.

**Figure supplement 2.** Variance in inter-region phase coherences of brain patterns.

(*Figure 3F*). We then repeated such computation for all subjects and all conditions (*Figure 3G*) and performed statistics on these distributions (shown in *Table 4A*). The slope was substantially heightened during and after stimulation with the cathodal montage compared to before stimulation. In addition, during stimulation and post-stimulation, the cathodal montage was associated with a significantly higher slope coefficient compared to the anodal montage. Consistent with our previous observations, the stimulation with the anodal montage had a negligible influence on this metric compared to the pre-stimulation condition. These results were not affected by the choice of the number of $k$ clusters (*Figure 3—figure supplement 1*), and similar results were obtained when the full analysis was repeated for $k = 4$ (*Figure 3—figure supplement 2*), as suggested by the IPVC (*Figure 2—figure supplement 1*). Further investigation of the variances in inter-region phase coherences of brain patterns, presented in *Figure 3—figure supplement 3*, revealed two main findings. First, all the patterns exhibited some degree of lateral symmetry. Second, except for the pattern with the

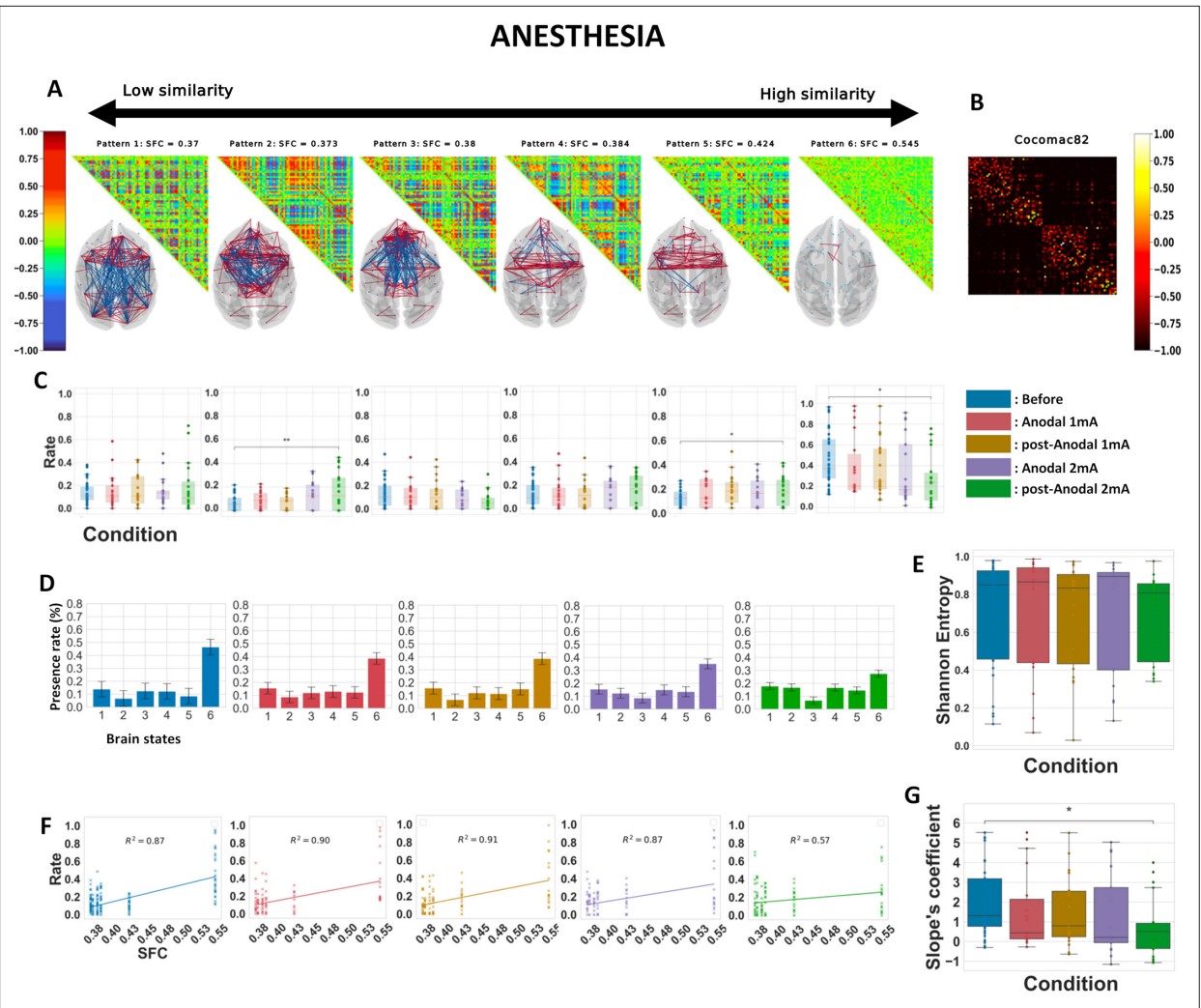

**Figure 6.** Anodal 2 mA transcranial direct current stimulation (tDCS) of the prefrontal cortex (PFC) modifies cortical brain dynamics and reduces structure–function coupling in anesthetized animals. Similar analyses to those in *Figures 3 and 5* (panels **A–G**) are presented here for *k* = 6, considering all the five anesthesia conditions. Before: before stimulation. Anodal 1 mA and post-Anodal 1 mA: during and after anodal tDCS of the PFC (F4/O1 or F3/O2) delivered at an intensity of 1 mA. Anodal 2 mA and post-Anodal 2 mA: during and after anodal tDCS of the PFC (F4/O1 or F3/O2 montage) delivered at an intensity of 2 mA. Asterisks indicate statistically significant differences between conditions (*: $p < 0.05$; **: $p < 0.01$).

The online version of this article includes the following figure supplement(s) for figure 6:

**Figure supplement 1.** Slope and Shannon entropy in anesthesia conditions.

**Figure supplement 2.** Dynamical functional connectivity analysis of the anesthesia dataset with *k* = 9 numbers of clusters.

**Figure supplement 3.** Variance in inter-region phase coherences of brain patterns.

highest SFC, most patterns displayed high heterogeneity in their coherence variances and striking inter-pattern differences. These observations reflect both the segmentation of distinct functional networks across patterns and a topological organization within the patterns themselves: some regions showed a broader spectrum of synchrony with the rest of the brain, while others exhibited narrower distributions of coherence variances. For instance, unlike other brain patterns, pattern 5 was characterized by a high coherence variance in the frontal premotor areas and low variance in the occipital cortex, whereas pattern 3 had a high variance in the frontal and orbitofrontal regions. In addition, we performed the main analyses separately for the two monkeys, explored the inter-condition variability (*Supplementary file 2*), and computed classical measures of functional connectivity such as average FC matrices and functional graph properties (modularity, efficiency, and density) of the visited FC states (*Supplementary file 3*). The separate analyses showed that the changes in slope and Shannon entropy were substantially more pronounced in one of the two monkeys, corroborating some of the

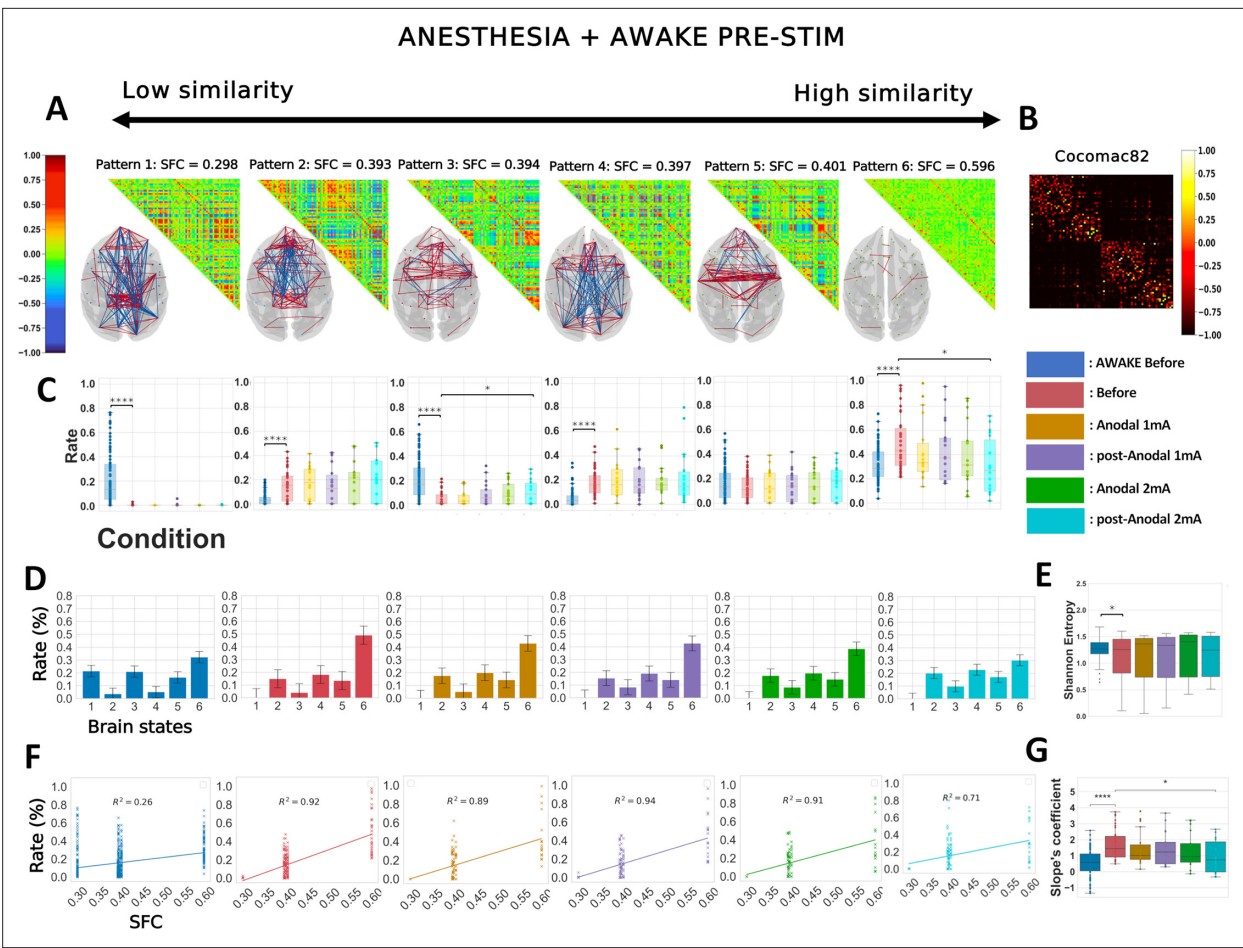

**Figure 7.** The effects of 2 mA anodal prefrontal transcranial direct current stimulation (tDCS) on anesthesia brain dynamics resist contrasting with awake dynamics. Similar analyses to those in *Figure 6* (panels **A-G**) for $k = 6$ are presented here, considering all the five anesthesia conditions plus the awake pre-stimulation condition. Awake before: before stimulation in the awake state. Before: before stimulation, under anesthesia. Anodal 1 mA and post-Anodal 1 mA: during and after anodal tDCS of the prefrontal cortex (PFC) at 1 mA intensity. Anodal 2 mA and post-Anodal 2 mA: during and after anodal tDCS of the PFC at 2 mA intensity. Asterisks indicate statistically significant differences between conditions (*: $p < 0.05$; ****: $p < 0.0001$).

The online version of this article includes the following figure supplement(s) for figure 7:

**Figure supplement 1.** Slope and Shannon entropy in anesthesia conditions contrasted with awake pre-stimulation condition.

**Figure supplement 2.** Variance in inter-region phase coherences of brain patterns.

effects captured in the ANOVA tests. In contrast, classical FC metrics did not show significant differences across stimulation conditions, highlighting the value of dynamic FC metrics to capture the neuromodulatory effects of tDCS.

Next, we analyzed the succession of brain states through the prism of a Markov chain. Both anodal and cathodal stimulations produced significant changes in probability transitions (*Figure 4*). The corresponding statistics are shown in *Table 5*. Strikingly, cathodal stimulation induced much greater transition probabilities from brain states 1 and 2 to brain state 6 (*Figure 4A*). Conversely, transitions from brain states 2 and 3, from 5 to 2, and 6 and 5 were significantly reduced by cathodal stimulation. Other changes were observed only after cathodal stimulation, including an increase in the probability of transitioning from brain states 5–6 and a probability decrease of transitioning from states 6–3. Ultimately, cathodal PFC tDCS strongly promoted transitions to brain state 6 and weakly inhibited some transitions leaving this state. As for anodal stimulation, it significantly favored the transitions from brain states 3 to 5 and from 2 to 3 and reduced the transitions from brain states 2 to 4 and 1 to 5 (*Figure 4B*). While these effects were not sustained post-stimulation, the transition from brain states 2 to 6 was significantly perturbed post-stimulation. Overall, anodal PFC tDCS weakly altered only a few transitions. Our results also show that the Markov chain transition probabilities are affected

**Table 4.** Statistical analyses.

Awake experiments.

| Brain state # | Compared conditions | Mean (SD) | df | t-value | Adjusted p-value |
|---|---|---|---|---|---|
| \multicolumn| T-test independent sample, Bonferroni correction, comparing occurrence of brain states | | | | |
| | Cathodal | 0.50 (0.15) | | | |
| 6 | Before | 0.37 (0.15) | 95 | 3 | 0.013 |
| | Post-Cathodal | 0.51 (0.18) | | | |
| 6 | Before | 0.37 (0.15) | 96 | 3.3 | 0.0052 |
| | Anodal | 0.33 (0.14) | | | |
| 6 | Cathodal | 0.50 (0.15) | 51 | 3.7 | 0.002 |
| | Post-Anodal | 0.34 (0.18) | | | |
| 6 | Post-Cathodal | 0.51 (0.18) | 53 | 3 | 0.016 |
| | Post-Cathodal | 0.028 (0.035) | | | |
| 5 | Before | 0.14 (0.14) | 96 | 3 | 0.012 |
| | Anodal | 0.16 (0.12) | | | |
| 3 | Cathodal | 0.038 (0.046) | 51 | 3.7 | 0.0022 |

| Brain state # | | Sum of squares | df | f-value | p-value |
|---|---|---|---|---|---|
| Three-way ANOVA type II | | | | | |
| | C(Montage) | 1.8e−14 | 2 | 3.7e−13 | 0.99 |
| | **C(Time)** | **4.2** | **2** | **84** | **<0.001** |
| | C(Animal) | NaN | 1 | NaN | NaN |
| | **C(Montage):C(Time)** | **1.1** | **4** | **11** | **<0.001** |
| | C(Montage):C(Animal) | 6.9e−4 | 2 | 0.014 | 0.91 |
| | C(Time):C(Animal) | 4.6e−5 | 2 | 9.2e−04 | 0.97 |
| | **C(Montage):C(Time):C(Animal)** | **0.99** | **4** | **9.9** | **<0.001** |
| 6 | Residual | 4.5 | 180 | NaN | |
| | C(Montage) | 1.7e−14 | 2 | 6.5e−13 | 0.99 |
| | C(Time) | 0.023 | 2 | 0.87 | 0.35 |
| | C(Animal) | NaN | 1 | NaN | NaN |
| | **C(Montage):C(Time)** | **0.17** | **4** | **3.2** | **0.04** |
| | **C(Montage):C(Animal)** | **0.24** | **2** | **9.1** | **0.0029** |
| | C(Time):C(Animal) | 4.30E−04 | 2 | 0.016 | 0.89 |
| | C(Montage):C(Time):C(Animal) | 0.14 | 4 | 2.8 | 0.066 |
| 5 | Residual | 2.3 | 180 | NaN | |

*Table 4 continued on next page*

*Table 4 continued*

| Brain state # | | Sum of squares | df | f-value | p-value |
|---|---|---|---|---|---|
| | C(Montage) | 2.70E−14 | 2 | 1.20E−12 | 0.99 |
| | **C(Time)** | **0.26** | **2** | **12** | **<0.001** |
| | C(Animal) | NaN | 1 | NaN | NaN |
| | **C(Montage):C(Time)** | **0.27** | **4** | **6.1** | **0.0026** |
| | C(Montage):C(Animal) | 0.039 | 2 | 1.8 | 0.18 |
| | C(Time):C(Animal) | 0.028 | 2 | 1.3 | 0.25 |
| | C(Montage):C(Time):C(Animal) | 0.12 | 4 | 2.6 | 0.07 |
| 3 | Residual | 1.9 | 180 | NaN | |

| Compared conditions | Mean (SD) | df | t-value | Adjusted p-value |
|---|---|---|---|---|
| *T*-test independent sample, Bonferroni correction, comparing non-normalized Shannon entropies | | | | |
| Cathodal | 1.1 (0.20) | | | |
| Before | 1.3 (0.23) | 95 | 2.5 | 0.037 |
| Post-Cathodal | 1.1 (0.22) | | | |
| Before | 1.3 (0.23) | 96 | 2.5 | 0.04 |
| Anodal | 1.3 (0.23) | | | |
| Cathodal | 1.1 (0.20) | 51 | 2.6 | 0.035 |

| Compared conditions | Mean (SD) | df | t-value | Adjusted p-value |
|---|---|---|---|---|
| *T*-test independent sample, Bonferroni correction, comparing slopes' coefficients | | | | |
| Cathodal | 1.9 (0.85) | | | |
| Before | 1.0 (0.86) | 95 | 2.9 | 0.018 |
| Post-Cathodal | 2.0 (1.0) | | | |
| Before | 1.0 (0.86) | 96 | 3.4 | 0.0043 |
| Anodal | 1.0 (0.80) | | | |
| Cathodal | 1.9 (0.85) | 51 | 3.5 | 0.0038 |
| Post-Cathodal | 2.0 (1.0) | | | |
| Post-Anodal | 1.0 (1.1) | 53 | 3 | 0.014 |

Values in bold indicate statistically significant difference (p < 0.05).

differently during and after tDCS. Some changes were observed only during ongoing current delivery, while others were specific to the post-stimulation condition. Importantly, the fact that both categories of changes were found in cathodal and anodal experiments suggests a general phenomenon.

## Contrasting the effects of prefrontal tDCS on the dynamic connectivity patterns in wakeful macaques to unconscious dynamics

Dynamic connectivity configurations associated with consciousness have previously been identified by analyzing data from conscious and unconscious states together (*Barttfeld et al., 2015*; *Uhrig et al., 2018*; *Demertzi et al., 2019*; *Castro et al., 2024b*). We therefore repeated our analysis of the data from all five awake experimental conditions pooled with the anesthesia pre-stimulation condition (*Figure 5*). All significant statistics are shown in *Table 6*.

The six resulting brain patterns showed somewhat different trends in connectivity arrangements compared to the patterns that emerged from the analysis of the awake data alone. While patterns 1 and 2 closely matched the previously obtained awake patterns 1 and 2, the others showed distinct

**Table 5.** Markov transition *T*-test independent* samples with Bonferroni correction.

| Transition | Compared conditions | Mean (SD) | df | t-value | Adjusted p-value |
|---|---|---|---|---|---|
| | Before | 0.24 (0.18) | | | |
| 2 → 4 | Anodal | 0.16 (0.23) | | 2.4 | 0.019 |
| | Before | 0.056 (0.13) | 118 | | |
| 3 → 5 | Anodal | 0.15 (0.19) | | 3.1 | 0.0074 |
| | Before | 0.34 (0.05) | | | |
| 2 → 6 | Post-Anodal | 0.47 (0.24) | | 4.7 | 0.0018 |
| | Before | 0.24 (0.11) | | | |
| 6 → 5 | Cathodal | 0.15 (0.16) | | 2.7 | 0.043 |
| | Before | 0.17 (0.11) | | | |
| 5 → 2 | Cathodal | 0.10 (0.12) | | 2.2 | 0.029 |
| | Before | 0.19 (0.17) | 95 | | |
| 2 → 3 | Cathodal | 0.08 (0.11) | | 2.4 | 0.01 |
| | Before | 0.56 (0.11) | | | |
| 1 → 6 | Cathodal | 0.73 (0.08) | | 5.7 | 0.002 |
| | Before | 0.35 (0.18) | | | |
| 2 → 6 | Cathodal | 0.66 (0.32) | | 3.6 | <0.001 |
| | Before | 0.15 (0.10) | | | |
| 6 → 3 | Post-Cathodal | 0.08 (0.11) | | 2.3 | 0.024 |
| | Before | 0.46 (0.15) | 96 | | |
| 5 → 6 | Post-Cathodal | 0.67 (0.23) | | 4.8 | <0.001 |

*Certain transitions, including transitions from brain states 4 and 5 to brain state 6, were affected both during and after cathodal stimulation. Such transitions were favored during and post-stimulation but did not pass the bootstrap and/or the ANOVA tests.

connectivity configurations (*Figure 5A*). As expected, both the awake and the anesthesia pre-stimulation conditions exhibited significant differences in the rates at which the six brain patterns occurred (*Figure 5C, D*). Pattern 1 was present in the awake pre-stimulation condition but virtually absent in the anesthesia pre-stimulation condition, whereas patterns 5 and 6 were predominant under anesthesia. These findings are consistent with previous dynamic rs-fMRI studies in macaques and humans (*Barttfeld et al., 2015*; *Uhrig et al., 2018*; *Demertzi et al., 2019*). The cathodal montage again produced strong perturbation effects. Compared to pre-stimulation, pattern 4, characterized by short-range frontal coherences, occurred for a significantly shorter time during and after stimulation. Pattern 6, the so-called anatomical pattern, showed differences between conditions, consistent with our previous analysis of the awake conditions. As previously observed, cathodal tDCS increased the occurrence rate of pattern 6, while anodal stimulation had no effect. Therefore, consistent with the prior analysis of the data taken from the awake conditions alone, the cathodal stimulation reduced the incidence of the pattern with a predominant frontal coherence (here, pattern 4), while it increased the incidence of the 'anatomical' pattern (pattern 6).

The Shannon entropy was found to be much lower in anesthetized monkeys than in awake animals (*Figure 5E*). This finding is consistent with previous studies in humans, regardless of whether unconsciousness was due to DOCs, deep sleep, or anesthesia (*Demertzi et al., 2019*; *Castro et al., 2024b*). While anodal stimulation had no effect, prefrontal cathodal stimulation significantly diminished Shannon entropy, similar to the previous analysis (*Figure 3E*). Consistent with our previous results, the cathodal tDCS induced a direct and lasting effect that was captured in our SFC-frequency analysis, where the states with high SFC were more present compared to those with low SFC (*Figure 5F, G*). These perturbations were noticeable when considering each macaque, and the effects observed

**Table 6.** Awake conditions + anesthesia pre-stimulation condition.

| Brain state # | Compared conditions | Mean (SD) | df | t-value | Adjusted p-value |
|---|---|---|---|---|---|
| *T*-test independent samples with Bonferroni correction | | | | | |
| | Cathodal | 0.027 (0.036) | | | |
| | Before | 0.13 (0.15) | 95 | 2.6 | 0.019 |
| | Post-Cathodal | 0.023 (0.037) | | | |
| 4 | Before | 0.13 (0.15) | 96 | 2.8 | 0.011 |
| | Anes Before | 0.54 (0.23) | | | |
| | Before | 0.35 (0.14) | 114 | 5.2 | <0.001 |
| | Cathodal | 0.47 (0.16) | | | |
| | Before | 0.35 (0.14) | 95 | 2.9 | 0.021 |
| | Post-Cathodal | 0.48 (0.17) | | | |
| | Before | 0.35 (0.14) | 96 | 3.1 | 0.0098 |
| | Cathodal | 0.47 (0.16) | | | |
| 6 | Anodal | 0.32 (0.14) | 51 | 3.4 | 0.0049 |

Two-way* ANOVA type II

| Brain state # | | Sum of squares | df | f-value | p-value |
|---|---|---|---|---|---|
| | **C(Montage)** | **0.27** | **2** | **7.5** | **0.0065** |
| | C(Time) | 0.094 | 2 | 2.7 | 0.1 |
| | **C(Montage):C(Time)** | **0.37** | **4** | **5.3** | **0.0056** |
| 4 | Residual | 3.9 | 219 | NaN | |
| | **C(Montage)** | **4.3** | **2** | **61** | **<0.001** |
| | **C(Time)** | **0.63** | **2** | **8.9** | **0.0032** |
| | **C(Montage):C(Time)** | **1.1** | **4** | **7.9** | **<0.001** |
| 6 | Residual | 7.7 | 219 | NaN | |

*T*-test independent sample, Bonferroni correction, comparing normalized Shannon entropies

| Compared conditions | Mean (SD) | df | t-value | Adjusted p-value |
|---|---|---|---|---|
| Before | 0.73 (0.12) | | | |
| Post-Cathodal | 0.65 (0.098) | 95 | 2.3 | 0.049 |
| Before | 0.73 (0.12) | | | |
| Anes Before | 0.51 (0.20) | 114 | 7 | <0.001 |

*T*-test independent sample, Bonferroni correction, comparing slopes' coefficients

| Compared conditions | Mean (SD) | df | t-value | Adjusted p-value |
|---|---|---|---|---|
| Cathodal | 1.3 (0.80) | | | |
| Before | 0.69 (0.72) | 95 | 2.8 | 0.014 |
| Post-Cathodal | 1.3 (0.85) | | | |
| Before | 0.69 (0.72) | 96 | 3 | 0.003 |
| Cathodal | 1.3 (0.80) | | | |
| Anodal | 0.55 (0.74) | 51 | 3 | 0.0072 |
| Post-Cathodal | 1.3 (0.85) | | | |
| Post-Anodal | 0.55 (0.74) | 53 | 2.6 | 0.023 |

Values in bold indicate statistically significant differences (p < 0.05).

*Due to the monkeys from the anesthesia datasets being different animals than those from the awake dataset, including the animal specialness as independent variable resulted in completely NaN three-way ANOVA results. Hence, we removed said variable to at least verify the effect of montage type and time.

during and post-cathodal stimulation were strong enough to alter the slopes. Compared to pre-stimulation, the changes observed during and post-stimulation were significant and had a similar tendency: an increase in the SFC-slope's coefficient. Finally, these differences were specific to the cathodal montage, and the comparison between the two types of electrode arrangements again revealed statistically significant differences. These results remained robust regardless of the number of *k* clusters (*Figure 5—figure supplement 1*). The variance in inter-regional phase coherence across brain patterns showed notably that pattern 4, in contrast to most other patterns, was characterized by a high variance in frontal premotor areas and a low variance in the occipital cortex (*Figure 5—figure supplement 2*).

## Effects of prefrontal tDCS on the dynamic connectivity patterns in anesthetized macaques

Due to the challenging nature of the experimental procedures under anesthesia, we limited the investigations to only one stimulation modality. We chose to deliver anodal stimulation to provide new information on the effects of tDCS under anesthesia as a model for DOCs and to increase the cortical excitability of the PFC in an attempt to wake up the sedated monkeys. We also adopted a slightly different experimental design, as we aimed to determine the effects of anodal tDCS of the PFC with increasing intensity. We acquired fMRI data before, during, and after tDCS delivered at an intensity of 1 mA, followed by a second block of stimulation at 2 mA (*Figure 1B*). Representative structural images showing electrode placements on the head of the two anesthetized monkeys are shown in *Figure 1—figure supplement 1B*. The complete set of structural images can be found in *Supplementary file 1*. A total of 136 runs were acquired in two macaque monkeys. We also continuously monitored vital parameters related to hemodynamics, ventilation, and temperature (*Table 3*). The dynamic signal coordination analysis was applied to the collected data. All significant statistics are shown in *Table 7*.

Compared to the patterns observed from the analysis of the awake conditions alone (*Figure 3A*), the patterns generated from this experimental setup showed different organizations of interconnectivity between regions, with patterns 2, 3, 4, and 5 displaying primarily frontal/prefrontal coherences (*Figure 6A*). Interestingly, in the 2 mA post-stimulation condition, brain states 2 and 5 presented an increase in their presence rates compared to pre-stimulation, while brain state 6 showed a decrease (*Figure 6C*). Plotting the frequency of pattern visits across conditions further illustrated the notable reduction in the prevalence of brain state 6, the higher occurrence rate of brain states 2 and 5, as well as a tendency for the effects of the stimulation to gradually rise as a function of stimulation intensity (*Figure 6D*). Comparisons of Shannon entropy were non-significant (*Figure 6E*). However, for the 2 mA post-stimulation condition, the slope coefficient was statistically reduced compared to pre-stimulation (*Figure 6F, G*). These results remained robust regardless of the number of *k* clusters (*Figure 6—figure supplement 1*). While several human studies have reported that 1 mA transcranial stimulation induces aftereffects (e.g., *Jamil et al., 2017*; *Jamil et al., 2020*; *Monte-Silva et al., 2010*), the differences between the 1 mA post-stimulation and baseline conditions were not significant in our analyses. However, it is still possible that the 1 mA stimulation produced some effects below the threshold of significance that may contribute to the changes observed during and after the 2 mA stimulation.

Because the IPVC suggested that nine brain patterns would be optimal to reach the maximum IPVC for the anesthesia dataset (*Figure 2—figure supplement 1*), we repeated the complete analysis for *k* = 9 (*Figure 6—figure supplement 2*). Using *k* = 9, we found even more perturbations in brain dynamics by tDCS than with *k* = 6. First, the 'anatomical' brain state's occurrence rate was significantly lower both during and following 2 mA stimulation. Additionally, brain states 5 and 8 (which resemble brain states 2 and 5 in the analysis with *k* = 6 in *Figure 6A*) showed an increase in their occurrence during and following stimulation at 2 mA. Finally, the slope coefficient was significantly decreased during and after 2 mA stimulation compared to before. No effect on the Shannon entropy was observed. The variance in inter-region phase coherences of the brain patterns is displayed in *Figure 6—figure supplement 3* and showed a striking heterogeneity between the patterns. For example, pattern 5 had a low overall variance (except in the frontal cortex), while pattern 1 was the only pattern with a high variance in the occipital cortex. Analyses of the two monkeys separately showed that the changes in slope and Shannon entropy were bigger in one of the two monkeys but went in the same direction

**Table 7.** Experiments under anesthesia.

| Brain state # | Compared conditions | Mean (SD) | df | t-value | Adjusted p-value |
|---|---|---|---|---|---|
| *T*-test independent samples with Bonferroni correction | | | | | |
| | Before | 0.067 (0.070) | | | |
| 2 | Post-Anodal 2 mA | 0.17 (0.17) | 103 | 3.1 | 0.0035 |
| | Before | 0.084 (0.069) | | | |
| 5 | Post-Anodal 2 mA | 0.15 (0.12) | | 2.2 | 0.03 |
| | Before | 0.46 (0.25) | | | |
| 6 | Post-Anodal 2mA | 0.27 (0.24) | | 3.1 | <0.001 |

| Brain state # | | Sum of squares | df | f-value | p-value |
|---|---|---|---|---|---|
| Three-way ANOVA type II | | | | | |
| | C(Montage) | −6.7e−17 | 2 | −6.3e−15 | 1.0 |
| | C(Time) | 1.7e−15 | 2 | 1.7e+13 | 1.0 |
| | **C(Animal)** | **0.45** | **1** | **88** | **<0.001** |
| | **C(Montage):C(Time)** | **0.18** | **4** | **8.9** | **<0.001** |
| | C(Montage):C(Animal) | 1.90e−17 | 2 | 1.8e−15 | 1.0 |
| | C(Time):C(Animal) | −4.90e−18 | 2 | −4.7.16 | 1.0 |
| | C(Montage):C(Time):C(Animal) | 0.083 | 4.0 | 4.0 | 0.0047 |
| 2 | Residual | 0.49 | 95 | NaN | |
| | C(Montage) | −8.50e−16 | 2 | −7.30e−14 | 1.0 |
| | C(Time) | −1.10e−16 | 2 | −9.20e−15 | 1.0 |
| | **C(Animal)** | **0.62** | **1** | **10.6** | **<0.001** |
| | **C(Montage):C(Time)** | **0.12** | **4** | **5.3** | **<0.001** |
| | C(Montage):C(Animal) | 1.00E−16 | 2 | 8.90E−15 | 1.0 |
| | C(Time):C(Animal) | 5.9–16 | 2 | 5.00E−14 | 1.0 |
| | C(Montage):C(Time):C(Animal) | 0.051 | 4 | 2.2 | 0.078 |
| 5 | Residual | 0.56 | 95 | NaN | |
| | C(Montage) | 1.90E−15 | 2 | −4.20E−14 | 1 |
| | C(Time) | −2.30E−16 | 2 | −5.10E−15 | 1 |
| | **C(Animal)** | **5.3** | **1** | **23.6** | **<0.001** |
| | **C(Montage):C(Time)** | **0.77** | **4** | **8.5** | **<0.001** |
| | C(Montage):C(Animal) | 2.90E−16 | 2 | 6.40E−15 | 1 |
| | C(Time):C(Animal) | 4.90E−15 | 2 | 8.90E−14 | 1 |
| | C(Montage):C(Time):C(Animal) | 0.11 | 4 | 1.2 | 0.32 |
| 6 | Residual | 2.1 | 95 | NaN | |

| Compared conditions | Mean (SD) | df | t-value | Adjusted p-value |
|---|---|---|---|---|
| *T*-test independent sample, Bonferroni correction, comparing slope's coefficients | | | | |
| Before | 1.5 (1.8) | | | |
| Post-Anodal 2 mA | 0.7 (1.5) | 51 | 2.6 | 0.025 |

Values in bold indicate statistically significant differences (p < 0.05).

(*Supplementary file 2*), while classical FC metrics still did not noticeably capture any changes or statistical differences between the different stimulation conditions (*Supplementary file 4*).

Next, we contrasted the data from all anesthesia conditions with the awake pre-stimulation condition, keeping $k = 6$ for the $k$-means clustering (*Figure 7*). As expected, all the patterns except pattern 5 showed significantly different occurrence rates between the awake and anesthesia pre-stimulation conditions (*Figure 7C*), consistent with the previous contrasted analysis (*Figure 5C*) and the literature (*Barttfeld et al., 2015*; *Uhrig et al., 2018*; *Demertzi et al., 2019*). Compared to pre-stimulation, brain state 3, a high positive frontal coherence pattern, was visited more often, and brain state 6 less frequently in the 2 mA post-stimulation condition. The slope coefficient was statistically reduced compared to pre-stimulation (*Figure 7G*), regardless of the number of $k$ clusters (*Figure 7—figure supplement 1*). These results show that 2 mA anodal PFC stimulation can reconfigure unconscious brain dynamics under anesthesia, albeit it could not restore the full range of characteristics of awake cortical dynamics, such as the visit of brain pattern 1. In line with these findings, these modulations were not accompanied by any behavioral signs of awakening in the animals. Nevertheless, the stimulation did influence some vital parameters (heart rate, systolic and diastolic blood pressure) (*Table 3*), as previously reported (*Gu et al., 2022*; *Rodrigues et al., 2022*). The variance in inter-region phase coherences of brain patterns is displayed in *Figure 6—figure supplement 2*.

Lastly, we studied how the Markov chain transitions between brain states across stimulation conditions under anesthesia. However, during and following stimulation, at both 1 and 2 mA, not a single transition passed the bootstrapping method, even when the confidence level was reduced to 90%. Thus, we could not interpret any possible changes in transition probability since those empirical transitions can't be considered specific nor temporally independent. As such, we do not show this analysis.

## Discussion

The dynamic nature of brain activity and connectivity in resting-state fMRI has attracted considerable interest over the past years. In this study, we demonstrate that tDCS perturbs dynamic brain patterns in both awake and deeply sedated macaques. We also demonstrate that the dynamic cortical connectivity arrangements previously described as typifying consciousness and unconsciousness can be non-invasively influenced by the application of a low-intensity direct current via transcranial stimulation. Our findings are consistent with the hypothesis that tDCS directly affects brain dynamics. In the following, we discuss the implications and potential mechanisms associated with the dynamic changes induced by tDCS.

### Cathodal prefrontal tDCS strongly perturbed cortical brain dynamics in awake macaques

In awake animals, our analysis showed that cathodal stimulation of the PFC markedly alters the relative prevalence of cortical signal coordination patterns (*Figures 3 and 5*). We found that the 'anatomical' pattern (pattern 6) was the most sensitive to cathodal stimulation. Our observations that the prevalence of this pattern and the transitions toward it are greatly enhanced by cathodal stimulation in a conscious wakeful context are both unexpected and surprising. One explanation might be that this pattern is related to transient drops in vigilance levels and that cathodal stimulation might enhance such a phenomenon. Contamination of fMRI data by transient periods of microsleep has been shown to occur frequently in awake resting-state experiments (*Tagliazucchi and Laufs, 2014*). Previous work suggested that tDCS over prefrontal or frontal regions can influence arousal, sleepiness, or sleep quality (*Frase et al., 2016*; *Annarumma et al., 2018*). For instance, cathodal bifrontal tDCS delivered immediately before sleep has been found to increase total sleep time and decrease cortical arousal in healthy volunteers (*Frase et al., 2016*). However, in our experiments, the stimulated animals showed no tendency to drowsiness or any sudden restlessness in an attempt to resist sleep. Furthermore, the animals were not resting but engaged in a fixation task throughout the fMRI data acquisition. It is also worth noting that they achieved a similar success rate in the cathodal stimulation condition compared to the pre-stimulation and anodal stimulation conditions (*Table 2*). Therefore, our observations do not appear to be consistent with an increase in drowsiness or a reduction in vigilance resulting from cathodal stimulation.

Alternatively, other types of mental processes might be enhanced by cathodal tDCS, such as the tendency to mind-wander, to experience mind-blanking, or to enter a hypnotic state. A seminal study demonstrated that anodal PFC stimulation can significantly enhance the tendency to mind-wander in healthy adults (*Axelrod et al., 2015*). However, subsequent efforts to replicate these findings yielded conflicting results (*Coulborn and Fernández-Espejo, 2022*; *Nawani et al., 2023*). Mind-blanking is a state where the mind is alert but not actively engaged in a specific cognitive task. Reports of mind-blanking were recently found to correlate with a pattern of global positive-phase coherence rather than the pattern that most closely resembles anatomy (*Mortaheb et al., 2022*). This finding does not support an enhancement of mind-blanking by cathodal stimulation of the PFC in our study.

Finally, cathodal left DLPFC tDCS was demonstrated to enhance hypnotizability by 15% and to alter some dimensions of consciousness, significantly increasing attention, absorption, and time sense but decreasing self-awareness and memory (*Perri et al., 2022*; *Perri and Di Filippo, 2023*). While our observations of the monkey's high fixation task performance would be consistent with increased attention or absorption, the demonstration of a modulatory effect of cathodal tDCS on one or other of these specific mental processes or dimensions of consciousness will require further investigation in humans using relevant behavioral measures, cognitive tasks, and brain activity tracking.

## Anodal prefrontal tDCS perturbed cortical brain dynamics, especially during deep anesthesia

In awake macaques, we found that the anodal montage significantly disturbed some transition probabilities between brain states (*Figure 4C, D*). Anodal stimulation tended to decrease the slope's coefficient (*Figures 3G and 5G*) and to increase the occurrence rate of brain state 1 (*Figures 3C, D and 5C, D*). Although small, these changes are consistent with recent literature suggesting that anodal PFC tDCS, when delivered during the daytime, can positively influence levels of vigilance and sleepiness (*Annarumma et al., 2018*). For example, this type of stimulation has been described to increase beta and decrease delta powers in a wakeful context, thus indicating enhanced vigilance (*Wirth et al., 2011*; *Maeoka et al., 2012*; *Dai et al., 2022*). Bifrontal anodal stimulation increased both objective and subjective vigilance and reduced daytime sleepiness in a patient with organic hyperinsomnia following reanimation (*Frase et al., 2015*). Moreover, in sleep-deprived adults, bifrontal anodal tDCS also reduced EEG correlates of physiological sleepiness (*Alfonsi et al., 2023*). Anodal left DLPFC tDCS performed better than caffeine in preventing the reduction in vigilance and alleviating subjective ratings of fatigue and drowsiness in sleep-deprived individuals (*McIntire et al., 2014*). Finally, such a stimulation affected total sleep time, shortening it by about 25 min, when administered just before sleep in healthy volunteers (*Frase et al., 2016*).

While we observed modest effects of anodal PFC tDCS in awake animals, the stimulation produced significant changes in cortical brain dynamics under anesthesia and reduced the SFC, suggesting a state in-between wakefulness and anesthesia (*Figures 6 and 7*). Therefore, although our protocol did not succeed in awakening the sedated animals, our results support the ability of prefrontal tDCS to affect brain dynamic patterns related to consciousness in the context of an anesthesia-induced loss of consciousness. We hypothesize that the molecular effect of the anesthetic drug was stronger than the electrical effect at an intensity of 2 mA, which could explain the lack of behavioral effects. However, a similar protocol could succeed in awakening macaques sedated with a lower dose of propofol or in reversing unconsciousness due to deep propofol anesthesia with a higher stimulation intensity. In this case, the effects may reach a threshold where we get as much disturbance from pharmacology as physical intervention, thus resulting in an observable behavioral effect of tDCS. This idea is supported by the fact that we observed a greater effect on brain dynamics at 2 mA compared to 1 mA. Interestingly, a recent study in humans showed that tDCS delivered at 4 mA intensity can robustly enhance the effects of stimulation compared to 2 mA and is safe, well-tolerated, and without adverse effects (*Nitsche and Bikson, 2017*; *Hsu et al., 2023*). Therefore, our results encourage further investigation using anodal prefrontal tDCS at an intensity of 4 mA to non-invasively attempt to reverse the effects of deep propofol anesthesia. Our findings also support the use of tDCS to promote rapid recovery from general anesthesia in humans (*Kato and Solt, 2021*) and suggest that a single anodal prefrontal stimulation at the end of the anesthesia protocol may be effective.

From another clinical perspective, our results demonstrating that 2 mA anodal PFC tDCS decreased the SFC and modified the dynamic repertoire of brain patterns during anesthesia (*Figures 6 and 7*)

are consistent with the beneficial effects of such stimulation in DOC patients (*Thibaut et al., 2014*; *Angelakis et al., 2014*; *Thibaut et al., 2017*; *Zhang et al., 2017*; *Martens et al., 2018*; *Cavinato et al., 2019*; *Wu et al., 2019*; *Hermann et al., 2020*; *Peng et al., 2022*; *Thibaut et al., 2023*). Although some clinical trials investigated the effects of stimulating other brain regions, such as the motor cortex (*Martens et al., 2019*; *Straudi et al., 2019*) or the parietal cortex (*Huang et al., 2017*; *Guo et al., 2019*; *Zhang et al., 2022*; *Wan et al., 2023*; *Wang et al., 2020*), the DLPFC appears to be the most effective target for patients with a minimally conscious state (*Liu et al., 2023*). In terms of neuromodulatory effects in DOC patients, DLPFC tDCS has been reported to increase global excitability (*Bai et al., 2017*), increase the P300 amplitude (*Zhang et al., 2017*; *Hermann et al., 2020*), improve the fronto-parietal coherence in the theta band (*Bai et al., 2018*), enhance the putative EEG markers of consciousness (*Bai et al., 2018*; *Hermann et al., 2020*), and reduce the incidence of slow waves in the resting state (*Mensen et al., 2020*). Our findings further support the PFC as a relevant target for modulating consciousness level and align with growing evidence showing that the PFC plays a key role in conscious access networks (*Mashour et al., 2022*; *Panagiotaropoulos, 2024*). Nevertheless, we hypothesize that other brain targets for tDCS may be of interest for consciousness restoration, potentially using multi-channel tDCS (*Havlík et al., 2023*). Among transcranial electrical stimulation techniques, tDCS has the great advantage of facilitating either excitation or inhibition of brain regions, depending on the polarity of the stimulation (*Sdoia et al., 2019*). Exploiting this advantage, they investigated the causal involvement of the DLPFC in conscious access to a visual stimulus during an attentional blink paradigm. While conscious access was enhanced by anodal stimulation of the left DLPFC compared to sham stimulation, opposite effects were found with cathodal stimulation compared to sham over the same locus. Finally, this literature and our findings suggest that tDCS constitutes a non-invasive, reversible, and powerful tool for studying consciousness.

Changes in the state of consciousness are generally closely associated with changes in behavioral responsiveness, although some rare cases of dissociation have been described. Cognitive-motor dissociation (CMD) is a condition observed in patients with severe brain injury, characterized by behavior consistent with unresponsive wakefulness syndrome or a minimally conscious state minus (*Thibaut et al., 2019*). However, in these patients, specific cortical brain areas activate in response to mental imagery tasks (e.g., imagining playing tennis or returning home) in a manner indistinguishable from that of healthy controls, as shown through fMRI or EEG (*Thibaut et al., 2019*; *Owen et al., 2006*; *Monti et al., 2010*; *Bodien et al., 2024*). Thus, although CMD patients are behaviorally unresponsive, they demonstrate cognitive awareness that is not outwardly apparent. It is worth noting that both the SFC and the rate of the pattern closest to the anatomy were shown to be significantly reduced in unresponsive patients showing command following during mental imagery tasks compared to those who do not show command following *Demertzi et al., 2019*. These observations would be compatible with our findings in anesthetized macaques exposed to 2 mA anodal PFC tDCS. The richness of the brain dynamics would be recovered (at least partially, in our experiments), but not the behavior. This hypothesis also fits with a recent longitudinal fMRI study on patients recovering from coma (*Crone et al., 2020*). The researchers examined two groups of patients: one group consisted of individuals who were unconscious at the acute scanning session but regained consciousness and improved behavioral responsiveness a few months later, and the second group consisted of patients who were already conscious from the start and only improved behavioral responsiveness at follow-up. By comparing these two groups, the authors could distinguish between the recovery of consciousness and the recovery of behavioral responsiveness. They demonstrated that only initially conscious patients exhibited rich brain dynamics at baseline. In contrast, patients who were unconscious in the acute phase and later regained consciousness had poor baseline dynamics, which became more complex at follow-up. Complete recovery of both consciousness and responsiveness under general anesthesia is possible through electrical stimulation of the central thalamus (*Redinbaugh et al., 2020*; *Tasserie et al., 2022*).

From a mechanistic point of view, prefrontal tDCS may modulate arousal and improve the level of consciousness by acting on the recently proposed cortico-thalamic pathway of sleep–wake regulation or by acting directly on the ascending thalamocortical pathway, which originates in the brainstem (*Krone et al., 2017*). Depending on electrode positions and stimulation parameters, tDCS has been shown to modulate subcortical regions directly. For example, fronto-temporal tDCS has been reported to alter corticostriatal and cortico-thalamic rs-fMRI connectivity in healthy volunteers (*Polanía et al.,*

2012; *Dalong et al., 2020*). In vivo measurements in humans using intracerebral electrodes confirmed that the stimulation can generate a biologically relevant electric field at the subthalamic level or other deep brain structures (*Guidetti et al., 2022*; *Louviot et al., 2022*). Computational modeling of the electric field (*Saturnino et al., 2019*; *Yu et al., 2012*) generated by the specific fronto-occipital montage used in this study will likely yield valuable information regarding which pathway (cortico-thalamic or thalamocortical) may have been modulated. Furthermore, modulation of intrinsic networks (*Raichle et al., 2001*; *Boly et al., 2008*) is possible in light of our result and could also be investigated through an iCAP-based analysis (*Karahanoğlu and Van De Ville, 2015*; *Ensel et al., 2023*).

From a biophysical standpoint, previous studies showed that anodal stimulation leads to depolarization at the soma of neurons in rat hippocampal slices, while cathodal stimulation induces their hyperpolarization, especially when the current is aligned with the soma–apical axis (*Bikson et al., 2004*). These findings were corroborated in human subjects (*Nitsche and Paulus, 2000*), where anodal tDCS applied to the motor cortex increased the amplitude of motor-evoked potentials in response to transcranial magnetic stimulation, while cathodal M1 stimulation had the opposite effects, revealing an increase or decrease in M1 excitability, respectively. The magnitude of these effects was small (0.2 mV) (*Bikson et al., 2004*) but fell within the range of endogenous electric field interactions generated between cortical columns, where it can be greatly amplified by recurrent interactions (*Rebollo et al., 2021*). Interestingly, in the awake state, we observed that cathodal stimulation of the PFC induced a significant decrease in the frequencies of patterns exhibiting high frontal coherence (patterns 3 and 5), which is consistent with a reduction in PFC excitability under the cathodal electrode. In contrast, the stimulation with this montage tended to increase the frequency of patterns with a predominant parieto-occipital coherence (patterns 2 and 4), which is in line with an increase in excitability under the anodal electrode positioned over the occipital cortex (*Figure 3A, C*). Under anesthesia, 2 mA anodal tDCS of the PFC increased the frequencies of some patterns exhibiting a high frontal coherence (patterns 2 and 5) (*Figure 6A, C*), also supporting enhanced excitability under the anodal electrode in deep sedation. It is important to note that the effects of the neuromodulatory ascending systems on brain activation are known to be mediated both by neuronal depolarization and by inhibition of voltage and calcium-dependent conductance, which are responsible for spike-frequency adaptation in pyramidal cells (*Steriade and McCarley, 2013*; *McCormick, 1992*). Therefore, one possible explanation for the inability of anodal stimulation of the PFC to arouse macaques from anesthesia is that it caused depolarization of neurons without affecting spike-frequency adaptation and thus only partially activated the brain.

## Markov chain analysis revealed differential tDCS effects on transition probabilities during and after stimulation

Our results showed relatively similar effects during and after the stimulation in terms of brain pattern occurrence rates, Shannon entropy, and SFC (*Figures 3 and 5–7*). In contrast, Markov chain analysis revealed the existence of specific changes in transition probability induced by tDCS, some of which were found only during stimulation and others exclusively in the post-stimulation condition (*Figure 4*, *Table 4B*). Our results highlight that certain effects of tDCS on brain dynamics are specific to the time course of the stimulation and can be revealed using particular dynamic analyses such as the Markov chain transition.

In conclusion, our data provide experimental evidence that tDCS of the PFC modulates spontaneous brain activity whether the animals are wakeful and conscious or unconscious under general anesthesia. Our study offers a deeper understanding of how tDCS influences and alters brain dynamics, which is crucial given its growing applications in neuroscience and the management of various neurological conditions. Finally, our work suggests that by allowing a soft, non-invasive, and reversible perturbation of brain activity, tDCS may be a valuable tool for further investigations of the relationship between brain dynamics and states of consciousness, conscious processing, or vigilance.

## Materials and methods
### Animals
Four rhesus monkeys (*M. mulatta*, 18–20 years, 7.2–9.8 kg) were scanned. Two males were tested for the awake experiments (monkeys J and Y), and one female and one male were tested for experiments

under anesthesia (monkeys R and N). Our sample size is comparable to previous work in NHP using fMRI (*Milham et al., 2018*; *Yacoub et al., 2020*; *Uchimura et al., 2024*).

All procedures followed the European Convention for Animal Care (86-406) and the National Institutes of Health's Guide for the Care and Use of Laboratory Animals. Animal studies were approved by the Institutional Ethical Committee (CETEA protocol #16-040).

## Transcranial direct current stimulation

We applied tDCS using a battery-powered stimulator (1 × 1 tES device, Soterix medical) located outside the scanner room and a complete MR-compatible setup consisting of an MRI filter, carbon composition cables (Soterix medical). We used a 1 × 1 electrode montage with two carbon rubber electrodes (dimensions: 1.4 cm × 1.85 cm, 0.93 cm thick) inserted into Soterix HD-tES MRI electrode holders (base diameter: 25 mm; height: 10.5 mm), which are in contact with the scalp. These electrodes ($2.59$ cm$^2$) are smaller than conventional tDCS sponge electrodes (typically 25–35 cm$^2$). The electrode setups and experimental designs used in awake and anesthetized macaques are shown in *Figure 1*.

Electrode contact quality was checked before all stimulation sessions using the device's Pre-stim Tickle function (1 mA for 30 s). If a poor contact was detected, the electrode-scalp contact was readjusted, and conductive paste or gel was added to improve contact quality. Electrode contact quality was continuously monitored during the stimulation.

To reduce the likelihood of the monkeys experiencing discomfort at the electrode sites, the current was gradually increased from 0 mA to the target intensity over the first 30 s. The current was also gradually decreased to 0 mA over the last 30 s of each tDCS stimulation period. The tDCS device was turned off during the acquisition of the pre- and post-stimulation fMRI blocks to avoid any hypothetical biological effect of the residual current (*Fonteneau et al., 2019*), which we measured to be 0.023 mA. After the tDCS-fMRI scanning sessions, the monkeys' scalps were examined for redness, skin burns, or other lesions. None of the monkeys showed any side effects.

### Awake experiments

In the awake experiments, elastic bands with fasteners were used to secure the electrode holders. We used a conductive and adhesive paste (Ten20 conductive paste, Weaver and Company) to ensure consistent electrode-skin contact. The anode electrode was placed over the left PFC (F3) and the cathode electrode over the right occipital cortex (O2), or the reverse montage was employed, with the cathodal electrode placed over the right PFC (F4) and the anode over the left occipital cortex (O1) (*Figure 1A*). We used monkey EEG caps (EasyCap, 13 channels, Fp1, Fp2, F3, F4, T3, T4, P3, P4, O1, O2, Oz, reference electrode, ground electrode), based on the international 10–20 EEG system, to obtain F3, F4, O1, and O2 locations, where we placed the stimulation electrodes. The two montages were, respectively, abbreviated F4/O1 and O1/F4. tDCS was delivered for 20 min at a current intensity of 2 mA.

### Anesthesia experiments

For anesthesia experiments, the electrode holders were held in place with Monkey-2D caps (EasyCaps GmbH), using an EEG gel to obtain low impedance (One Step EEG gel, Germany) as the conductive medium. Monkey R was stimulated with an F3/O2 montage (anodal electrode over the left PFC and cathodal electrode over the right occipital cortex) and monkey N was stimulated with an F4/O1 montage (anodal electrode over the right PFC and cathodal electrode over the left occipital cortex) (*Figure 1B*). This different but still close placement of the electrodes between the two animals was dictated by constraints related to the location of the headpost (with the cement used to anchor the headpost spreading more or less asymmetrically) and the anatomy/specificity of each animal's head. Because of the small size of the monkey's head and because we did not use return electrodes to restrict the current flow (as is achieved with typical high-definition montages; *Datta et al., 2009*; *Alam et al., 2016*), we expected that tDCS stimulation with the two symmetrical montages would result in nearly equivalent electric fields across the monkey's head and produce roughly similar effects on brain activity. This would need to be confirmed by running an electric field simulation. However, computational electric field models have been developed for humans, and their use in NHPs is not straightforward due to anatomical specificities. Indeed, monkeys differ from humans in terms of brain size, shape

and cortical organization, skull thickness, tissue conductivities and the presence of muscles (*Lee et al., 2015*; *Datta et al., 2016*; *Mantell et al., 2023*). Modeling of EFs generated with the specific tDCS montages employed in this study will be performed in future work. Two 20-min blocks of anodal tDCS stimulation were applied consecutively in each scanning session, with a current intensity of 1 mA for the first block and 2 mA for the second block (*Figure 1B*).

## fMRI data acquisition

The monkeys were scanned on a 3-T horizontal scanner (Siemens Prisma Fit, Erlanger, Germany) with a custom-built single transmit-receiver surface coil. The parameters of the fMRI sequences were as follows: (1) functional scan: EPI, TR = 2400 ms, TE = 20 ms, 1.5 mm isotropic voxel size, and 111 brain volumes per run, (2) anatomical scan: MPRAGE, T1-weighted, TR = 2200 ms, TE = 3.18 ms, 0.80 mm isotropic voxel size, sagittal orientation. Before each scanning session, monocrystalline iron oxide nanoparticles (10 mg/kg, i.v.; MION, Feraheme, AMAG Pharmaceuticals, MA) were injected into the saphenous vein of the monkey to improve the contrast-to-noise ratio and the spatial selectivity of the MR signal changes (*Vanduffel et al., 2001*).

All fMRI data were acquired while the monkeys were engaged in a passive event-related auditory task, the local–global paradigm, which is based on local and global deviations from temporal regularities (*Bekinschtein et al., 2009*; *Uhrig et al., 2014*). The present paper does not address how tDCS perturbs cerebral responses to local and global deviants, which will be the subject of future work. fMRI data were acquired before tDCS stimulation, during tDCS stimulation (onset of fMRI acquisition starting after the ramp-up period and stopping before the ramp-down period), and after tDCS stimulation. For anesthesia experiments, a second round of stimulation/post-stimulation block was completed (see below). Four runs of 111 brain volumes were acquired during each condition (before tDCS stimulation, during tDCS stimulation, and after tDCS stimulation). Finally, all sessions ended up with an anatomical scan.

### Awake protocol

Monkeys were implanted with an MR-compatible headpost, trained to sit in a sphinx position in a primate chair, and to fixate within a 2 × 2° window centered on a red dot (0.35 × 0.35°) within a 'mock' MR bore before being scanned. fMRI data were acquired in awake macaques as previously described (*Uhrig et al., 2014*). The eye position was monitored at 120 Hz (Iscan Inc, MA, USA) and the fixation rate during the fMRI run was recorded (*Table 2*). To ensure that the results were not biased by fatigue or drowsiness due to the lengthy experimental procedure, only runs with a fixation rate >85% were included in the analysis. No physiological parameters were recorded in the awake state.

A total of 288 runs were acquired in the awake state: 136 runs before stimulation, 56 runs during anodal stimulation, 56 runs after anodal stimulation, 20 runs during cathodal stimulation, and 20 runs after cathodal stimulation (see *Table 1* for details of the number of runs acquired per monkey).

### Anesthesia protocol

The anesthesia protocol is described in detail in a previous study (*Tasserie et al., 2022*). Anesthesia was induced with an intramuscular injection of ketamine (10 mg/kg, Virbac, France) and dexmedetomidine (20 µg/kg, Ovion Pharma, USA) and maintained with a target-controlled infusion (TCI) (Alaris PK Syringe pump, CareFusion, CA, USA) of propofol (Panpharma Fresenius Kabi, France) using the 'Paedfusor' pharmacokinetic model (*Absalom and Kenny, 2005*) (Monkey R: TCI 5.8 µg/ml, Monkey N: TCI 4.3 µg/ml). Monkeys were intubated and mechanically ventilated. The physiological parameters (heart rate, non-invasive blood pressure, oxygen saturation, respiratory rate, end-tidal carbon dioxide, and skin temperature) were monitored (Maglife, Schiller, France) and recorded (*Table 3*). A muscle-blocking agent (cisatracurium, GlaxoSmithKline, France, 0.15 mg/kg, bolus i.v., followed by a continuous infusion rate of 0.18 mg/kg/hr) was used during all anesthesia fMRI sessions to avoid artifacts.

We referred to a clinical arousal scale at the beginning and end of each scanning session to characterize the behavior of monkeys in experiments under anesthesia. This scale comprises several indicators: an exploration of the environment, spontaneous movements, shaking/prodding, toe pinch, eye-opening, and corneal reflex (for more details, see *Uhrig et al., 2016*; *Tasserie et al., 2022*).

A total of 136 runs were acquired under anesthesia: 40 runs before stimulation, 24 runs during 1 mA anodal stimulation, 24 runs after 1 mA anodal stimulation, 24 runs during 2 mA anodal stimulation, and 24 runs after 2 mA anodal stimulation (see *Table 1* for details of the number of runs acquired per monkey).

## Statistical analysis of the physiological data

One-way ANOVA was used to compare the effect of the stimulation on mean heart rate, blood pressure, and other vital parameters under anesthesia (*Table 3*). Statistical analysis and multiple comparisons were performed using home-made MATLAB scripts (MathWorks, USA), with a statistical threshold of $p < 0.05$, Bonferroni-corrected.

## fMRI data analyses

### fMRI preprocessing

Functional images were reoriented, realigned, resampled (1 mm isotropic), smoothed (Gaussian kernel, 3 mm full width at half maximum), and rigidly co-registered to the anatomical template of the monkey MNI space (*Frey et al., 2011*) using custom-made scripts (*Uhrig et al., 2014*).

The global signal was regressed out from the images to remove any confounding effects due to physiological cha-nges (e.g., respiratory or cardiac variations). Voxel time series were filtered with low-pass (0.05 Hz cutoff) and high-pass (0.0025 Hz cutoff) filters and a zero-phase fast Fourier notch filter (0.03 Hz) to remove an artifactual pure frequency present in all the data (*Barttfeld et al., 2015*; *Uhrig et al., 2018*).

An additional cleaning step was performed to check the data quality after time series extraction, as described previously (*Signorelli et al., 2021*). The procedure was based on a trial-by-trial visual inspection by an expert in neuroimaging of the time series for all nodes, the Fourier transform of each signal, the functional connectivity for each subject, and the dynamical connectivity computed with phase correlation. For each dataset, visual inspection was first used to become familiar with the characteristics of the entire dataset: how the amplitude spectrum, time series, FC, and dynamic FC look. Subsequently, each trial was inspected again with a particular focus on two main types of potential artifacts. The first one corresponds to potential issues with the acquisition and is given by stereotyped sinusoidal oscillatory patterns without variation. The second one refers to potential head or other movement not fully corrected by our preprocessing procedure. This last artifact can be sometimes recognized by bursts or peaks of activity. Sinusoidal activity generates artificially high functional correlation and peak of frequencies in the amplitude spectrum plot. Uncorrected movements generate peaks of activity with high functional correlation and sections of high functional correlations in the dynamical FC matrix. If we observed any of these anomalies, we rejected the trial, opting to adopt a conservative policy. Trials were retained if the row signal showed no evidence of artifactual activity, the functional connectivity was coherent with the average and the dynamical connectivity showed consistent patterns over time.

Finally, a total of 295 runs were analyzed in the following conditions: awake state, 190 runs (before tDCS, 82 runs; during anodal tDCS, 38 runs; after anodal tDCS, 39 runs; during cathodal tDCS, 15 runs; after cathodal tDCS, 16 runs) and anesthesia, 105 runs (before tDCS, 34 runs; during 1 mA anodal tDCS, 18 runs; after 1 mA anodal tDCS, 18 runs; during 2 mM anodal tDCS, 17 runs; after 2 mA anodal tDCS, 18 runs).

## Anatomical parcellation and structural connectivity

Anatomical connectivity data were obtained from a recent macaque connectome generated by coupling diffusion MRI tractography with axonal tract tracing studies (*Bakker et al., 2012*; *Shen et al., 2019*). The macaque cortex was subdivided using the Regional Map parcellation method (*Kötter and Wanke, 2005*). The parcellation includes 82 cortical ROIs (41 ROIs per hemisphere). Structural (i.e., anatomical) connectivity data are represented as matrices in which the 82 cortical ROIs are plotted on the *x*- and *y*-axis. The CoCoMac connectivity matrix contains information about the strength of the connections between cortical areas, with each cell representing the connection between any two regions.

## Signals representation

Given a real-valued signal $x(t)$, there exists a unique analytic representation $\widetilde{x}(t)$, which is a complex signal, with the same Fourier transform as the real-valued signal strictly defined for positive frequencies. This analytic signal can be constructed from the real-valued signal using the Hilbert transform $H$:

$$\widetilde{x}(t) = x(t) + iH[x(t)], \tag{1}$$

$i$ being the squared root of −1, the imaginary unit. The motivations for using the analytic signal are that, given some real-value data (i.e., MION signals), one can determine two functions of time that provide more meaningful properties of the signal. For a narrowband signal that can be written as an amplitude-modulated low-pass signal $A(t)$ with carrier frequency expressed by $\varphi(t)$:

$$x(t) = A(t)\ cos[\varphi(t)]. \tag{2}$$

Then, if the Fourier transforms of $A(t)$ and $cos(\varphi(t))$ have separate supports, the analytic signal of a narrowband signal can be rewritten as the product of two meaningful components

$$\widetilde{x}(t) = A(t)\ e^{i\varphi(t)}, \tag{3}$$

where $A(t)$ is the instantaneous amplitude, $\varphi(t)$ the instantaneous phase obtained from the Hilbert transform $H[x(t)]$.

The narrower the bandwidth of the signal of interest, the better the Hilbert transform produces an analytic signal with a meaningful envelope and phase (*Glerean et al., 2012*). Adopting a band-pass filtered version of the MION time series $x(t)$ improves the separation between the phase and envelope spectra. The time series was $z$-scored, and subsequently, the Hilbert transform was computed to obtain the phase of each signal at each volume and each moment (TR, i.e., one-time sample).

## Phase-based dynamic functional coordination

A recurrent issue in fMRI studies dealing with dynamical analysis is the arbitrary choices in the time-windows length and their overlapping when capturing temporal oscillations through a sliding-window methodology (*Hindriks et al., 2016*; *Xie et al., 2019*; *Allen et al., 2014*; *Barttfeld et al., 2015*). Thus, phase-based dynamic functional coordination was preferred (*Alonso Martínez et al., 2020*; *Cabral et al., 2017*; *Vohryzek et al., 2020*; *Demertzi et al., 2019*). Analytic representations of signals were employed to derive a phase signal corresponding to the MION time series. We computed the instantaneous phase $\varphi_n(t)$ of the signals across all ROIs $n \in \{1, \ldots N\}$ for each TR $t \in 2, \ldots, T-1$. The first and last TR of each fMRI scan was excluded due to possible signal distortions induced by the Hilbert transform (*Bracewell, 2000*).

The instantaneous phase was computed using Euler's formula from the analytic signal, which was then 'enfolded' within the range of $-\pi$ to $\pi$, facilitating the calculation of inter-ROI phase differences. To obtain a whole-brain pattern of phase differences, the phase coherence between pairs of areas $k$ and $j$ at each time $t$, labeled $PC(k, j, t)$, was estimated using the pairwise phase coherence defined as the cosine of the angular difference:

$$PC(k, j, t) = cos(\varphi_k(t) - \varphi_j(t)). \tag{4}$$

When areas $k$ and $j$ have synchronized MION signals at time $t$, the phase coherence takes the value 1, and when areas $k$ and $j$ are in anti-phase at time $t$, their associated phase coherence is −1. All the intermediate cases lie in between the interval [−1,1]. This computation was repeated for all subjects. For each subject, the resulting phase coherence was a three-dimensional tensor with dimension $N * N * T$, where $N$ is the total number of regions in the parcellation (here 82 regions) and $T$ the total number of TR in each fMRI session. One can easily see the symmetry with respect to the spatial dimensions ($PC(k, j, t) = PC(j, k, t)$). Hence, only the triangular superior parts of the phase coherence

matrices were used for later computations, thus reducing the phase space's size to $\frac{N*(N-1)}{2}$ (the diagonal was omitted) and containing the useful available information.

### *K*-means clustering

To assess recurring coordination patterns among individuals in both datasets, a multistep methodology was employed. First, the scanning sessions were transformed into matrices, wherein one dimension represented instantaneous phase differences (feature space), as explained above, and the other dimension represented time. We concatenate all the vectorized triangular superior parts for all subjects at all times and from the experimental conditions, hence obtaining phase-spaces over time matrices, one for each condition. Each resulting matrix was subjected to the *k*-means clustering algorithm utilizing the L2 'Euclidean' distance metric. This resulted in a discrete set of *k* coordination patterns and their corresponding occurrence over time.

This process yielded *k* cluster centroids, which served as representatives of the recurring coordination patterns, accompanied by a label indicating the closest pattern for each instantaneous phase difference at every given TR.

The number of clusters was determined for a range $k = 3, \dots 10$, with 100 initializations and 200 iterations each, using the 'k-means++' initialization method, respectively, to ensure a maximum (or minimum) value for metrics used as a proxy for choosing the optimal *k*-number of clusters, and for the stability in the centroids obtained. To determine the most appropriate number of clusters, we employed the IPCV as the primary method, as we are mainly interested in the centroids' aspect, supplemented by the elbow method for confirmation.

The IPCV is computed by vectorizing the set of centroids for a given *k* and calculating the variance of Pearson correlations between each pair of centroids (*Demertzi et al., 2019*). The optimal *k*, indicating the number of clusters, is determined by identifying the maximum value of this metric. To validate this optimal k, we employed the elbow method, which is typically based on the within-cluster sum of squares differences. While the elbow was not always as distinctly observable as desired, it still often coincided with the proposed *k* from the IPCV.

### Structure–function correlation

To investigate the dependence of brain dynamics on the state of consciousness, we defined a measure of similarity between functional and structural connectivity. We resorted to a widely used group estimate of macaque anatomical connectivity provided by the CoCoMac 2.0 database. We computed the linear correlation coefficient between the entries of both matrices (*k*-means centroids and anatomical connectome) for each set of *k*-clusters. This was performed for each phase-based coordination pattern obtained using the *k*-means algorithm and was denoted SFC. We then reorganized the patterns' labels, their unique associated integer between 1 and *k*, such that the new 'first' pattern would be the pattern with the lowest SFC, and the '*k*th' pattern the one with the highest correlation to the anatomy (i.e., structure). Last, we computed the linear slope coefficient of the relationship between the occurrence rate of each centroid and the corresponding centroid/anatomical correlation (i.e., the SFC) to establish a possible tendency for functional states with higher (or lower) correlation to the anatomy.

### Brain states' presence rate distribution and Shannon entropy

For each condition, we computed the histogram associated with the total time of visits of each pattern. This histogram, after normalization, corresponds to the probability distribution of the patterns' occurrence. For each of the normalized histogram of occurrence, and each value of *k* in the *k*-means algorithm, we obtained the Shannon entropy (*Shannon, 1948*) as follows:

$$S = -\sum_i P_i log_2 P_i,$$ (5)

where $P_i$ is the probability (normalized histogram) of visit for each pattern in each brain state (condition).

### Markov chain

For every TR, the *k*-means algorithm assigns the index of the closest centroid to the corresponding empirical phase coherence matrix, resulting in a sequence of indexes (integers, between 1 and *k*).

We reckon the sequence of 'patterns visited' as a sequence of random variables that jumps from one pattern to another pattern with a certain probability given by the empirical frequencies in which these transitioning events happen in the sequence. From these empirical frequencies, we built a Markov chain, as done previously (*Demertzi et al., 2019*). More specifically, we counted the number of transitions between all pairs of patterns without considering self-transitions (which correspond to remaining in the same pattern) and normalized these counts to obtain a Markov transition probability matrix $P$.

## Statistical analysis

To assess the statistical significance of comparing different conditions and differences in occurrence rates, slopes' coefficients, and Shannon entropy, we performed independent *T*-tests with Bonferroni correction between the awake and anesthesia conditions before any stimulation and all their associated stimulation conditions and reported the associated p-values. We also performed three- or two-way ANOVA to confirm that the obtained results of comparing two conditions together (with corrections for multiple comparisons) are consistent with comparing the conditions given the independent variables at play: during/after stimulation, amperage/montage used, and the animal; and to evaluate possible interactions between said variables.

All these analyses are shown in table form (*Table 4*) following APA standards by reporting the means and standard deviations, the statistics (*t*- or *F*-values), the degrees of freedom and the p-values, all to two significant digits. For the ANOVA tests, we also report the sum of squares.

To ensure the reliability of the transition probabilities, we performed a bootstrap method that breaks the temporal dependencies in the chain of visited states. We randomized the order of brain states' appearance, thus conserving the overall statistics but breaking the temporal dependence between specific pairs of visited states. We performed this randomization 10,000 times per condition and computed as many transition matrices. We asserted that certain states' transitions truly reflected a stochastic process and were hence not due to chance by verifying if the empirical transition matrices were above the confidence interval defined by successively stricter confidence levels: 90%, 95%, and 99%. This also allows us to somewhat compensate for the animal-specific effects by discarding them. Finally, for the transitions that did pass at least the 90% confidence interval test, we compared, using independent *T*-tests with Bonferroni correction, the transition probabilities between conditions and reported the associated p-values.

## Acknowledgements

The research was supported by the Institut National de la Santé et de la Recherche Médicale (to GH and BJ), the Fondation pour la Recherche Médicale (FRM grant number ECO20160736100, to JT), FNRS Belgium (project MIS/VA – F.4523.21, to CMS), grant Embodied-Time (40011405, to CMS), Centre National de la Recherche Scientifique and the European Community (Human Brain Project, H2020-945538, to AD and RC), the Fondation Bettencourt Schueller (to BJ and GH), Fondation de France (to BJ), Human Brain Project (Corticity project FLAC-ERA JTC2017, to BJ), UVSQ (to BJ), Commissariat à l'Énergie Atomique (to BJ), Collège de France (to BJ).

## Additional information

### Funding

| Funder | Grant reference number | Author |
|---|---|---|
| Fondation pour la Recherche Médicale | ECO20160736100 | Jordy Tasserie |
| Fonds De La Recherche Scientifique - FNRS | F.4523.21 | Camilo Miguel Signorelli |
| Centre National de la Recherche Scientifique | H2020-945538 | Alain Destexhe Rodrigo Cofre |
| Fondation Bettencourt Schueller | | Guylaine Hoffner Béchir Jarraya |

| Funder | Grant reference number | Author |
| --- | --- | --- |
| Institut National de la Santé et de la Recherche Médicale | | Guylaine Hoffner<br>Béchir Jarraya |
| Fondation de France | | Béchir Jarraya |
| European Commission | FLAG-ERA JTC2017 | Béchir Jarraya |
| Université Paris-Saclay | | Béchir Jarraya |
| Commissariat à l'Énergie Atomique et aux Énergies Alternatives | | Béchir Jarraya |
| FNRS-Belgium | Grant Embodied-Time 40011405 | Camilo Miguel Signorelli |
| Université de Versailles Saint-Quentin-en-Yvelines | | Béchir Jarraya |
| Collège de France | | Béchir Jarraya |

The funders had no role in study design, data collection, and interpretation, or the decision to submit the work for publication.

## Author contributions

Guylaine Hoffner, Conceptualization, Data curation, Supervision, Investigation, Methodology, Writing – original draft, Writing – review and editing; Pablo Castro, Software, Formal analysis, Validation, Visualization, Writing – original draft; Lynn Uhrig, Conceptualization, Supervision, Investigation, Methodology, Writing – review and editing; Camilo Miguel Signorelli, Writing – review and editing, Preprocessing of the data; Morgan Dupont, Jordy Tasserie, Investigation; Alain Destexhe, Funding acquisition, Validation, Writing – review and editing; Rodrigo Cofre, Software, Formal analysis, Supervision, Funding acquisition, Validation, Visualization, Writing – review and editing; Jacobo Sitt, Conceptualization, Resources, Software, Formal analysis, Supervision, Validation, Visualization, Methodology, Writing – review and editing; Béchir Jarraya, Conceptualization, Resources, Software, Supervision, Funding acquisition, Validation, Investigation, Methodology, Project administration, Writing – review and editing

## Author ORCIDs

Guylaine Hoffner ⓘ https://orcid.org/0000-0001-7027-7919
Pablo Castro ⓘ https://orcid.org/0009-0005-7262-0916
Lynn Uhrig ⓘ http://orcid.org/0000-0002-2737-0197
Camilo Miguel Signorelli ⓘ http://orcid.org/0000-0002-2110-7646
Alain Destexhe ⓘ https://orcid.org/0000-0001-7405-0455
Rodrigo Cofre ⓘ http://orcid.org/0000-0002-7498-7122
Jacobo Sitt ⓘ https://orcid.org/0000-0002-3878-4846
Béchir Jarraya ⓘ https://orcid.org/0000-0003-0878-763X

## Ethics

All procedures followed the European Convention for Animal Care (86-406) and the National Institutes of Health's Guide for the Care and Use of Laboratory Animals. Animal studies were approved by the Institutional Ethical Committee (CETEA protocol #16-040).

Reviewer #2 (Public review): https://doi.org/10.7554/eLife.101688.3.sa1
Reviewer #3 (Public review): https://doi.org/10.7554/eLife.101688.3.sa2
Author response https://doi.org/10.7554/eLife.101688.3.sa3

# Additional files

## Supplementary files

Supplementary file 1. Structural images showing the position of the transcranial direct current

stimulation (tDCS) electrodes on the monkey's head across sessions. Sagittal, coronal, and transverse MRI sections, and corresponding skin reconstruction images showing the position of the prefrontal and occipital electrodes on the monkey's head for each MRI session (except for four sessions in which no anatomical scan was acquired). The two electrodes were accurately placed over the prefrontal cortex and the occipital cortex in a reproducible manner across sessions and between the two monkeys studied in each arousal state. In anesthesia experiments, the anodal electrode was placed over the dorsal prefrontal cortex, while the cathodal electrode was positioned over the parieto-occipital junction. In awake experiments, the prefrontal electrode was positioned over the dorsal prefrontal cortex/premotor cortex, while the occipital electrode was placed over the visual area 1. The position of the two electrodes differed slightly between the anesthetized and awake experiments due to different body positions (the prone position of the sedated monkeys prevented a more posterior position of the occipital electrode) and also due to the presence of a headpost on the head of the two monkeys in awake experiments (the monkeys we worked with in anesthesia experiments did not have a headpost).

Supplementary file 2. Complementary figures showing the analysis performed separately for the two monkeys reproducing the main findings. (A) Slope analysis performed for both monkeys (Y. and J.) separately in the only awake conditions. (B) Shannon entropy analysis performed for both monkeys (Y. and J.) separately in the only awake conditions. (C) Slope analysis performed for both monkeys (Y. and J.) separately in the only anesthesia conditions. (D) Shannon entropy analysis performed for both monkeys (Y. and J.) separately in the only anesthesia conditions.

Supplementary file 3. FC matrices changes and effects of transcranial direct current stimulation (tDCS) on classical functional graph properties in awake dataset. (A) Particular region-to-region functional links are depicted by their p-value significance when comparing pairs of conditions. (B) Modularity of average FC matrices shows no difference between conditions, (C) neither does it when computed for all the visited functional phase coherence states (for every time point, for all subjects), nor efficiency measure or density.

Supplementary file 4. FC matrices changes and effects of transcranial direct current stimulation (tDCS) on classical functional graph properties in anesthesia dataset. (A) Particular region-to-region functional links are depicted by their p-value significance when comparing pairs of conditions. Not a single functional connection was found to be statistically significant. (B) Modularity of average FC matrices shows no difference between conditions, (C) neither does it when computed for all the visited functional phase coherence states (for every time point, for all subjects), nor efficiency measure or density.

MDAR checklist

### Data availability

The raw fMRI data are available from the corresponding authors through academic collaboration. All these analyses were performed using customized codes available on an open Github repository for transparency, reproducibility, and open science (https://github.com/Krigsa/phase_coherence_kmeans, copy archived at *Castro et al., 2024a*). Known Python libraries such as add_state_annot (*Charlier et al., 2022*) or Scipy (*Virtanen et al., 2020*) were used to perform the statistical analysis notably.

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
