## [Editor Report · eLife Assessment]

This **valuable** study applies transcranial direct current stimulation (tCDS) to the prefrontal cortex of non-human primates during two states: (1) propofol-induced unconsciousness; and (2) wakeful performance of a fixation task. The analysis offers **incomplete** evidence to indicate that the effect of tDCS on brain dynamics, as recorded with functional magnetic resonance imaging, is contingent on the state of consciousness during which the stimulation is applied. The findings will be of interest to researchers interested in brain stimulation and consciousness.

---

## [Referee Report · Reviewer #2 (Public review)]

General comments

We thank the reviewers and editor for their thoughtful feedback. We are glad that the minor comments appear resolved. In this revision, we added subject-specific analyses, further FC comparisons, and clarified our rationale for stimulation parameters. We acknowledge that two concerns remain: (1) the 1 mA-2 mA sequence may introduce confounds, and (2) electric field modeling was not included due to technical limitations. We now explicitly note these as limitations in the manuscript and provide justification and discussion accordingly.

Major comments

R.2.1. For the anesthetized monkeys, the anode location differs between subjects, with the electrode positioned to stimulate the left DLFPC in monkey R and the right DLPFC in monkey N. The authors mention that this discrepancy does not result in significant differences in the electric field due to the monkeys' small head size. However, this is incorrect, as placing the anode on the left hemisphere would result in a much lower EF in the right DLPFC than placing the anode on the right side. Running an electric field simulation would confirm this. Additionally, the small electrode size suggested by the Easy cap configuration for NHP appears sufficient to stimulate the targeted regions focally. If this interpretation is correct, the authors should provide additional evidence to support their claim, such as a computational simulation of the EF distribution.

R.2.1 Authors' answer: We thank the Reviewer for the comments. First, regarding the reviewer's statement that placing the anode on the left hemisphere would result in a much lower EF in the right DLPFC than placing the anode on the right side, we would like to clarify that we did not use a typical 4 x 1 concentric ring high-definition setup (which consists of a small centre electrode surrounded by four return electrodes), but a two-electrode montage, with one electrode over the left or right PFC and the other one over the contralateral occipital cortex. According to EF modelling papers, a 4 x 1 high-definition setup would produce an EF that is focused and limited to the cortical area circumscribed by the ring of the return electrodes (Datta et al. 2009; Alam et al. 2016). Therefore, targeting the left or right DLPFC with a 4 x 1 setup would produce an EF confined to the targeted hemisphere of the PFC. In contrast, we expect the brain current flow generated with our 2-electrode setup to be broader, despite the small size of the electrodes, because there is no constraint from return electrodes. Thus, with our setup, the current is expected to flow between the PFC and the occipital cortex (see also our responses to comments R3.3., R.E.C.#2.1. and R.E.C.#2.2.).

Second, we would like to point out that in awake experiments, in which we stimulated the right PFC of both monkeys, there was no gross evidence of left or right asymmetry in the computed functional connectivity patterns (Figure 3A, Figure 3 - figure supplement 2A; Figure 5A). These results, showing that our stimulation montages did not induce asymmetric dynamic FC changes in NHPs, support the idea that our setups did not generate EFs that were spatially focused enough to alter brain activity in one hemisphere substantially more than the other.

Third, it is also worth noting that current evidence suggests that human brains are significantly more lateralized than those of macaques. Macaque monkeys have been found to have some degree of lateralized networks, but these are of lower complexity, and the lateralization is less pronounced and functionally organized than in humans. (Whey et al., 2014; Mantini et al., 2013). This suggests that, even if the stimulation were focal enough to stimulate the left or the right part of the PFC only, the behavioural effects would likely be similar.

Follow-up comment: Thank you for the detailed response and for referencing both experimental data and prior literature. While I appreciate the discussion on the lack of functional asymmetry and reduced lateralization in macaques, my original concern was about the physical distribution of the electric field (EF) due to different anode placements. Functional connectivity outcomes do not necessarily reflect EF symmetry, and without EF modeling, it's difficult to determine whether the stimulation affected both hemispheres equally. I understand the challenges of NHP-specific modeling, but even a simplified simulation or acknowledgment of this limitation in the manuscript would help clarify the interpretability of your results.

R.2.2. For the anesthetized monkeys, the authors applied 1 mA tDCS first, followed by 2 mA tDCS. A 20-minute stimulation duration of 1 mA tDCS is strong enough to produce after-effects that could influence the brain state during the 2 mA tDCS. This raises some concerns. Previous studies have shown that 1 mA tDCS can generate EF of over 1 V/m in the brain, and the effects of stimulation are sensitive to brain state (e.g., eye closed vs. eye open). How do the authors ensure that there are no after-effects from the 1 mA tDCS? This issue makes it challenging to directly compare the effects of 1 mA and 2 mA stimulation.

R.2.2 Authors' answer: We agree with the reviewer's comment that 1 mA tDCS may induce aftereffects, as has been observed in several human studies (e.g., Jamil et al. 2017, 2020). Although the differences between the 1 mA post-stimulation and baseline conditions were not significant in our analyses, it's still possible that the stimulation produced some effects below the threshold of significance that may contribute, albeit weakly, to the changes observed during

Follow-up comment: Thank you for the clarification and for acknowledging the potential for 1 mA after-effects. While I appreciate the authors' transparency and the amendment to the manuscript, I still find it important that the limitation be clearly stated in the Discussion section. The fact that 2 mA stimulation always followed 1 mA introduces a potential confound, making it difficult to attribute observed changes uniquely to 2 mA. If a counterbalanced design was not feasible, I would recommend explicitly noting this as a limitation in the interpretation of dose-dependent effects.

R.2.3. The occurrence rate of a specific structural-functional coupling pattern among random brain regions shows significant effects of tDCS. However, these results seem counterintuitive. It is generally understood that non-invasive brain stimulation tends to modulate functional connectivity rather than structural or structural-functional connectivity. How does the occurrence rate of structural-functional coupling patterns provide a more suitable measure of the effectiveness of tDCS than functional connectivity alone? I would recommend that the authors present the results based on functional connectivity itself. If there is no change in functional connectivity, the relevance of changes in structural-functional coupling might not translate into a meaningful alteration in brain function, making it unclear how significant this finding is without corresponding functional evidence.

R.2.3. Authors' answer: First of all, we would like to make it clear that the occurrence rate of patterns as a function of their SFC is not intended to be used or seen as a 'better' measure of the efficacy of tDCS. Instead, it is one aspect of the effects of tDCS on whole-brain functional cortical dynamics, obtained from refined measures (phase-coherences), that specifically addresses the coupling between structure and function. This type of analysis is further motivated by its increasing use in the literature due to its suspected relationship to wakefulness (e.g., (Barttfeld et al. 2015, Demertzi et al. 2019; Castro et al. 2023)). Also, in our analysis, the structure is kept constant: the connectivity matrix used to correlate the functional brain states is always the same (CoCoMac82). Thus, the influence of tDCS on the structure-function side can only be explained by modulating the functional aspects, as suggested by intuition and previous results.

Then, we agree with the reviewer that studying the functional changes induced by tDCS alone could be valuable. However, usual metrics used in FC analysis are usually done statistically: FC-states are either computed through averaging spatial correlations over time, then analyzed through graph-theoretical properties for instance (or by just directly computing the element-wise differences), or either by considering the properties of the different visited FC-states by computing spatial correlations over a sliding time-window, and then similar analysis can be done as previously explained. But these are static metrics, if the states visited are essentially the same (which is expected from non-invasive neuromodulations that haven't already demonstrated strong and/or characteristic impact), but the dynamical process of visiting said states changes, one would see no difference in that regard. As such, in the case of resting-state fMRI, differences in FCs are hard to interpret given that between-sessions within-condition differences are usually found with some degree of variance for the respective conditions. Trying then to interpret between-condition differences is quite tricky in the case of subtle modulations of the system's activity. On the other hand, more subtle differences can be captured by considering more detailed analysis, such as using phase-based methods like we did, by incorporating some statistical learning component with regard to the dynamicity of the system (supervised learning for instance like we did followed by temporal & transition-based methodology), and by adding some dimensions along which one will be able to give some interpretation to the analysis. In our case we were interested in characterizing resting-state differences between stimulation conditions, which have nuanced and subtle interactions with the biological system. As such, classical measures of differences between FC states are likely to not be refined and precise enough. In fact, we propose additional files investigating those classically used measures such as differences in average FC matrices, or changes in functional graph properties (like modularity, efficiency and density) of the visited FC states. These figures show that, for the first case, comparing region-to-region specific FCs provides very few statistically significant results. With respect to the second part, we show that virtually no differences are observed in the properties of the functional states visited. These results suggest, as expected, that the actual brain states visited across the different stimulation conditions are topologically quite similar, and that only very few region-specific pairwise functional connectivities are particularly modulated by specific tDCS montages while, on the other hand, the actual dynamical process dictating how the brain activity passes from one state to another is in fact being influenced as shown by the dynamical analysis presented in the main figures in a more apparent and meaningful way (in that it is dependent on the montage, somewhat consistent with regard to the post-stimulations conditions, and can be made sense of by considering the theoretical effect of near-anodal versus near-cathodal neuromodulatory effects).

Actions in the text: We have added new supplementary files showing the effects of the stimulations on FC matrices and on classical functional graph properties in awake and anesthesia datasets (Supplementary Files 3 & 4). We have added new sentences about these new analyses on the effects of the stimulations on FC matrices and on classical functional graph properties in the Results section:

Follow-up comment: Thank you for the detailed and comprehensive response. The clarification regarding the use of SFC dynamics and the additional analyses provided are convincing.

R2.4. The authors recorded data from only two monkeys, which may limit the investigation of the group effects of tDCS. As the number of scans for the second monkey in each consciousness condition is lower than that in the first monkey, there is a concern that the main effects might primarily reflect the data from a single monkey. I suggest that the authors should analyze the data for each monkey individually to determine if similar trends are observed in both subjects.

R.2.4. Authors' answer: We agree that the small number of subjects is a limitation of our study. However, we have already addressed these aspects by reporting statistical analyses that consider them, using linear models of such variables, and running them through ANOVA tests. In addition, we experimentally ensured that we recorded a relatively high number of sessions over a period of several years. Regardless, we agree that our study would benefit from further investigation into this matter. We have therefore prepared complementary figures showing the main analysis performed separately for the two monkeys as proposed, as well as further investigations into the inter-condition variability outmatching the inter-individual variability, itself being also outmatched by intra-individual changes.

Actions in the text: We have added a supplementary file showing the main analyses performed separately for the two monkeys (Supplementary File 2) and further investigations into the inter-condition variability (Supplementary Files 3 & 4). We have added new sentences about these analyses performed separately for the two monkeys in the Results section:

Follow-up comment: Thank you for addressing this concern and for providing the individual monkey analysis. The additional figures and statistical explanations are helpful and appreciated.

R2.5. Anodal tDCS was only applied to anesthetized monkeys, which limits the conclusion that the authors are aiming for. It raises questions about the conclusion regarding brain state dependency. To address this, it would be better to include the cathodal tDCS session for anesthetized monkeys. If cathodal tDCS changes the connectivity during anesthesia, it becomes difficult to argue that the effects of cathodal tDCS vary depending on the state of consciousness as discussed in this paper. On the other hand, if cathodal tDCS would not produce any changes, the conclusion would then focus on the relationship between the polarity of tDCS and consciousness. In that case, the authors could maintain their conclusion but might need to refine it to reflect this specific relationship more accurately.

R.2.5. Authors' answer: We agree with the reviewer that it would have been interesting to investigate the effects of cathodal tDCS in anesthetized monkeys. However, due to the challenging nature of the experimental procedures under anesthesia, we had to limit the investigations to only one stimulation modality. We chose to deliver anodal stimulation because, from a translational point of view, we aimed to provide new information on the effects of tDCS under anesthesia as a model for disorders of consciousness. It also made much more sense to increase the cortical excitability of the prefrontal cortex in an attempt to wake up the sedated monkeys rather than doing the opposite.

Actions in the text: We have added a new sentence in the Results section:

"Due to the challenging nature of the experimental procedures under anesthesia, we limited the investigations to only one stimulation modality. We chose to deliver anodal stimulation to provide new information on the effects of tDCS under anesthesia as a model for disorders of consciousness and to increase the cortical excitability of the PFC in an attempt to wake up the sedated monkeys."

Follow-up comment: Thank you for clarifying the rationale behind applying only anodal stimulation under anesthesia. While I appreciate the experimental constraints and the translational motivation, I would still encourage the authors to explicitly acknowledge in the Discussion that the absence of a cathodal condition under anesthesia limits the ability to dissociate polarity-specific effects from state-dependent effects. This clarification would help temper the conclusions and better reflect the scope of the current dataset.

---

## [Referee Report · Reviewer #3 (Public review)]

Summary:

This study used transcranial direct current stimulation administered using small 'high definition' electrodes to modulate neural activity within the non-human primate prefrontal cortex during both wakefulness and anaesthesia. Functional magnetic resonance imaging (fMRI) was used to assess neuromodulatory effects of stimulation. The authors report on modification of brain dynamics during and following anodal and cathodal stimulation during wakefulness and following anodal stimulation at two intensities (1 mA, 2 mA) during anaesthesia. This study provides some support that prefrontal direct current stimulation can alter neural activity patterns across wakefulness and sedation in monkeys.

Strengths and Weaknesses:

A key strength of this work is the use of fMRI-based methods to track changes in brain activity with good spatial precision. Another strength is the exploration of stimulation effects across wakefulness and sedation, which has the potential to provide novel information on the impact of electrical stimulation across states of consciousness. The authors should be commended for undertaking this challenging and important work.

The lack of a sham stimulation condition is a limitation of the study, as it somewhat restricts the certainty with which the results can be attributed to the active stimulation as opposed to other external factors such as drowsiness or fatigue as a result of the experimental procedure? Nevertheless, I acknowledge the demanding nature of performing this work in non-human primates and that only runs with high fixation rates were included, which should have helped reduce any fatigue-related effects.

In the anaesthesia condition, the authors investigated the effects of two intensities of stimulation (1 mA and 2 mA). However, it is possible that the initial 1 mA stimulation block might have caused some level of plasticity-related changes in neural activity that could have potentially interfered with the following 2 mA block due to the lack of a sufficient wash-out period. This potentially limits the findings from the 2 mA block as they cannot be interpreted as completely separate to the initial 1 mA stimulation period, given that they were administered consecutively. However, I do acknowledge the author's point that differences between the 1 mA post-stimulation and baseline conditions were not significantly different, which provides some evidence against this possibility.

The different electrode placement for the two anaesthetised monkeys (i.e., Monkey R: F3/O2 montage, Monkey N: F4/O1 montage) is potentially problematic, as it might have resulted in stimulation over different brain regions. Electric field models of brain current flow for the monkeys would really be needed to understand with reasonable certainty, however, I recognise that these models are generally designed for human studies making their integration into non-human primate studies challenging.

Finally, the sample size is obviously small. However, I realise this is often a limitation in non-human primate research, which can be both expensive and labour intensive.

Assessment:

This manuscript presents some novel insights into the effects of transcranial direct current stimulation on functional brain dynamics in awake and anaesthetised monkeys using MRI-based connectivity indices. Overall, the study presents several interesting and potentially impactful findings regarding stimulation-induced changes in brain activity. There are some limitations, such as the small sample size, lack of a sham stimulation control, and lack of electric field models, which, if included, would have, in my view, further helped improve the rigour of the study. Nevertheless, the manuscript presents several important findings, which warrant further analysis in future work.

---

## [Author Response]

The following is the authors’ response to the original reviews.

**Reviewer #1 (Public review):**
Summary:In this work, the authors apply TDCS to awake and anesthetized macaques to determine the effect of this modality on dynamic connectivity measured by fMRI. The question is to understand the extent to which TDCS can influence conscious or unconscious states. Their target was the PFC. During the conscious states, the animals were executing a fixation task. Unconsciousness was achieved by administering a constant infusion of propofol and a continuous infusion of the muscle relaxant cisatracurium. They observed the animals while awake receiving anodal or cathodal hd-TDCS applied to the PFC. During the cathodal stimulation, they found disruption of functional connectivity patterns, enhanced structure-function correlations, a decrease in Shannon entropy, and a transition towards patterns that were more commonly anatomically based. In contrast under propofol anesthesia anodal hd-TDCS stimulation appreciably altered the brain connectivity patterns and decreased the correlation between structure and function. The PFC stimulations altered patterns associated with consciousness as well as those associated with unconsciousness.Strengths:The authors carefully executed a set of very challenging experiments that involved applying tDCS in awake and anesthetized non-human primates while conducting functional imaging.We thank the Reviewer for summarising our study and for his appreciation of the highly challenging experiments we performed.Weaknesses:The authors show that tDCS can alter functional connectivity measured by fMRI but they do not make clear what their studies teach the reader about the effects of tDCS on the brain during different states of consciousness. No important finding is stated contrary to what is stated in the abstract. It is also not clear what the work teaches us about how tDCS works nor is it clear what are the "clinical implications for disorders of consciousness." The deep anesthesia is akin to being in a state of coma. This was not discussed.While the authors have executed a set of technically challenging experiments, it is not clear what they teach us about how tDCS works, normal brain neurophysiology, or brain pathological states such as disorders of consciousness.

We thank the reviewer for his comments. We agree that we could better highlight the value and implications of our work, and we take this opportunity to improve our manuscript according to the suggestions.

Actions in the text: We have added several new paragraphs in the Discussion section, considering these comments and other related remarks from the Reviewing Editor (see below our answer to the first comment of the Reviewing Editor: REC#1).

**Reviewer #2 (Public review):**
General comments:The authors investigated the effects of tDCS on brain dynamics in awake and anesthetized monkeys using functional MRI. They claim that cathodal tDCS disrupts the functional connectivity pattern in awake monkeys while anodal tDCS alters brain patterns in anesthetized monkeys. This study offers valuable insight into how brain states can influence the outcomes of noninvasive brain stimulation. However, there are several aspects of the methods and results sections that should be improved to clarify the findings.

We thank the Reviewer for the summary and appreciation of our study.

Major commentsFor the anesthetized monkeys, the anode location differs between subjects, with the electrode positioned to stimulate the left DLFPC in monkey R and the right DLPFC in monkey N. The authors mention that this discrepancy does not result in significant differences in the electric field due to the monkeys' small head size. However, this is incorrect, as placing the anode on the left hemisphere would result in a much lower EF in the right DLPFC than placing the anode on the right side. Running an electric field simulation would confirm this. Additionally, the small electrode size suggested by the Easy cap configuration for NHP appears sufficient to stimulate the targeted regions focally. If this interpretation is correct, the authors should provide additional evidence to support their claim, such as a computational simulation of the EF distribution.

We thank the Reviewer for the comments. First, regarding the reviewer’s statement that placing the anode on the left hemisphere would result in a much lower EF in the right DLPFC than placing the anode on the right side, we would like to clarify that we did not use a typical 4 x 1 concentric ring high-definition setup (which consists of a small centre electrode surrounded by four return electrodes), but a two-electrode montage, with one electrode over the left or right PFC and the other one over the contralateral occipital cortex. According to EF modelling papers, a 4 x 1 high-definition setup would produce an EF that is focused and limited to the cortical area circumscribed by the ring of the return electrodes (Datta et al. 2009; Alam et al. 2016). Therefore, targeting the left or right DLPFC with a 4 x 1 setup would produce an EF confined to the targeted hemisphere of the PFC. In contrast, we expect the brain current flow generated with our 2-electrode setup to be broader, despite the small size of the electrodes, because there is no constraint from return electrodes. Thus, with our setup, the current is expected to flow between the PFC and the occipital cortex (see also our responses to comments R3.3., R.E.C.#2.1. and R.E.C.#2.2.).

Second, we would like to point out that in awake experiments, in which we stimulated the right PFC of both monkeys, there was no gross evidence of left or right asymmetry in the computed functional connectivity patterns (Figure 3A, Figure 3 - figure supplement 2A; Figure 5A). These results, showing that our stimulation montages did not induce asymmetric dynamic FC changes in NHPs, support the idea that our setups did not generate EFs that were spatially focused enough to alter brain activity in one hemisphere substantially more than the other.

Third, it is also worth noting that current evidence suggests that human brains are significantly more lateralized than those of macaques. Macaque monkeys have been found to have some degree of lateralized networks, but these are of lower complexity, and the lateralization is less pronounced and functionally organized than in humans. (Whey et al., 2014; Mantini et al., 2013). This suggests that, even if the stimulation were focal enough to stimulate the left or the right part of the PFC only, the behavioural effects would likely be similar.

We strongly agree with the reviewer that conducting an EF simulation would be valuable to confirm our expectations and to gain a comprehensive view of the characteristics of the EFs generated with our different setups in NHPs. However, the challenge is in the fact that EF computational models have been developed for humans, and their use in NHPs is not straightforward due to significant anatomical differences. For example, macaque monkeys are distinct from humans in terms of brain size, shape and cortical organisation, skull thickness, and the presence of muscles, as well as different tissue conductivities (Lee et al. 2015; Datta et al.2016; Mantell et al. 2023). We plan to address this in future work.

Actions in the text: In the Materials and Methods section, we have modified the sentence: “Because of the small size of the monkey's head and because we did not use return electrodes to restrict the current flow (as is achieved with typical high-definition montages (Datta et al. 2009; Alam et al. 2016)), we expected that tDCS stimulation with the two symmetrical montages would result in nearly equivalent electric fields across the monkey’s head and produce roughly similar effects on brain activity.”

We also added a new sentence about EF simulation:

“This would need to be confirmed by running an electric field simulation. However, computational electric field models have been developed for humans, and their use in NHPs is not straightforward due to anatomical specificities. Indeed, monkeys differ from humans in terms of brain size, shape and cortical organization, skull thickness, tissue conductivities and the presence of muscles (Lee et al. 2015; Datta et al. 2016; Mantell et al. 2023). Modelling of EFs generated with the specific tDCS montages employed in this study will be performed in future work.”

For the anesthetized monkeys, the authors applied 1 mA tDCS first, followed by 2 mA tDCS. A 20-minute stimulation duration of 1 mA tDCS is strong enough to produce after-effects that could influence the brain state during the 2 mA tDCS. This raises some concerns. Previous studies have shown that 1 mA tDCS can generate EF of over 1 V/m in the brain, and the effects of stimulation are sensitive to brain state (e.g., eye closed vs. eye open). How do the authors ensure that there are no after-effects from the 1 mA tDCS? This issue makes it challenging to directly compare the effects of 1 mA and 2 mA stimulation.

We agree with the reviewer's comment that 1 mA tDCS may induce aftereffects, as has been observed in several human studies (e.g., Jamil et al. 2017, 2020). Although the differences between the 1 mA post-stimulation and baseline conditions were not significant in our analyses, it's still possible that the stimulation produced some effects below the threshold of significance that may contribute, albeit weakly, to the changes observed during and after 2 mA stimulation. We have, therefore, amended the paper in line with the reviewer's comments.

Actions in the text: We have added the following text in the Result section:

“While several human studies have reported that 1 mA transcranial stimulation induces aftereffects (e.g., Jamil et al. 2017, 2020; Monte-Silva et al. 2010), the differences between the 1 mA post-stimulation and baseline conditions were not significant in our analyses. However, it is still possible that the 1 mA stimulation produced some effects below the threshold of significance that may contribute to the changes observed during and after the 2 mA stimulation.”

The occurrence rate of a specific structural-functional coupling pattern among random brain regions shows significant effects of tDCS. However, these results seem counterintuitive. It is generally understood that noninvasive brain stimulation tends to modulate functional connectivity rather than structural or structural-functional connectivity. How does the occurrence rate of structural-functional coupling patterns provide a more suitable measure of the effectiveness of tDCS than functional connectivity alone? I would recommend that the authors present the results based on functional connectivity itself. If there is no change in functional connectivity, the relevance of changes in structural-functional coupling might not translate into a meaningful alteration in brain function, making it unclear how significant this finding is without corresponding functional evidence.

First, of all, we would like to make it clear that the occurrence rate of patterns as a function of their SFC is not intended to be used or seen as a ‘better’ measure of the efficacy of tDCS. Instead, it is one aspect of the effects of tDCS on whole-brain functional cortical dynamics, obtained from refined measures (phase-coherences), that specifically addresses the coupling between structure and function. This type of analysis is further motivated by its increasing use in the literature due to its suspected relationship to wakefulness (e.g., (Barttfeld et al. 2015, Demertzi et al. 2019; Castro et al. 2023)). Also, in our analysis, the structure is kept constant: the connectivity matrix used to correlate the functional brain states is always the same (CoCoMac82). Thus, the influence of tDCS on the structure-function side can only be explained by modulating the functional aspects, as suggested by intuition and previous results.

Then, we agree with the reviewer that studying the functional changes induced by tDCS alone could be valuable. However, usual metrics used in FC analysis are usually done statistically: FC-states are either computed through averaging spatial correlations over time, then analyzed through graph-theoretical properties for instance (or by just directly computing the element-wise differences), or either by considering the properties of the different visited FC-states by computing spatial correlations over a sliding time-window, and then similar analysis can be done as previously explained. But these are static metrics, if the states visited are essentially the same (which is expected from non-invasive neuromodulations that haven’t already demonstrated strong and/or characteristic impact), but the dynamical process of visiting said states changes, one would see no difference in that regard. As such, in the case of resting-state fMRI, differences in FCs are hard to interpret given that between-sessions within-condition differences are usually found with some degree of variance for the respective conditions. Trying then to interpret between-condition differences is quite tricky in the case of subtle modulations of the system’s activity. On the other hand, more subtle differences can be captured by considering more detailed analysis, such as using phase-based methods like we did, by incorporating some statistical learning component with regard to the dynamicity of the system (supervised learning for instance like we did followed by temporal & transition-based methodology), and by adding some dimensions along which one will be able to give some interpretation to the analysis. In our case we were interested in characterizing resting-state differences between stimulation conditions, which have nuanced and subtle interactions with the biological system.

As such, classical measures of differences between FC states are likely to not be refined and precise enough. In fact, we propose additional files investigating those classically used measures such as differences in average FC matrices, or changes in functional graph properties (like modularity, efficiency and density) of the visited FC states. These figures show that, for the first case, comparing region-to-region specific FCs provides very few statistically significant results. With respect to the second part, we show that virtually no differences are observed in the properties of the functional states visited.

These results suggest, as expected, that the actual brain states visited across the different stimulation conditions are topologically quite similar, and that only very few region-specific pairwise functional connectivities are particularly modulated by specific tDCS montages while, on the other hand, the actual dynamical process dictating how the brain activity passes from one state to another is in fact being influenced as shown by the dynamical analysis presented in the main figures in a more apparent and meaningful way (in that it is dependent on the montage, somewhat consistent with regard to the post-stimulations conditions, and can be made sense of by considering the theoretical effect of near-anodal versus near-cathodal neuromodulatory effects).

Actions in the text: We have added new supplementary files showing the effects of the stimulations on FC matrices and on classical functional graph properties in awake and anesthesia datasets (Supplementary Files 3 & 4).

We have added new sentences about these new analyses on the effects of the stimulations on FC matrices and on classical functional graph properties in the Results section:

“In addition, we performed the main analyses separately for the two monkeys, explored the inter-condition variability (Supplementary File 2), and computed classical measures of functional connectivity such as average FC matrices and functional graph properties (modularity, efficiency and density) of the visited FC states (Supplementary File 3).... In contrast, classical FC metrics did not show significant differences across stimulation conditions, highlighting the value of dynamic FC metrics to capture the neuromodulatory effects of tDCS.”

“Analyses of the two monkeys separately showed that the changes in slope and Shannon entropy were bigger in one of the two monkeys but went in the same direction (Supplementary File 2), while classical FC metrics did not capture any statistical differences between the different stimulation conditions (Supplementary File 3).”

The authors recorded data from only two monkeys, which may limit the investigation of the group effects of tDCS. As the number of scans for the second monkey in each consciousness condition is lower than that in the first monkey, there is a concern that the main effects might primarily reflect the data from a single monkey. I suggest that the authors should analyze the data for each monkey individually to determine if similar trends are observed in both subjects.

We agree that the small number of subjects is a limitation of our study. However, we have already addressed these aspects by reporting statistical analyses that consider them, using linear models of such variables, and running them through ANOVA tests. In addition, we experimentally ensured that we recorded a relatively high number of sessions over a period of several years. Regardless, we agree that our study would benefit from further investigation into this matter. We have therefore prepared complementary figures showing the main analysis performed separately for the two monkeys as proposed, as well as further investigations into the inter-condition variability outmatching the inter-individual variability, itself being also outmatched by intra-individual changes.

Actions in the text: We have added a supplementary file showing the main analyses performed separately for the two monkeys (Supplementary File 2) and further investigations into the inter-condition variability (Supplementary Files 3 & 4).

We have added new sentences about these analyses performed separately for the two monkeys in the Results section:

“In addition, we performed the main analyses separately for the two monkeys, explored the inter-condition variability (Supplementary File 2), and computed classical measures of functional connectivity such as average FC matrices and functional graph properties (modularity, efficiency and density) of the visited FC states (Supplementary File 3). The separate analyses showed that the changes in slope and Shannon entropy were substantially more pronounced in one of the two monkeys, corroborating some of the effects captured in the ANOVA tests.”

“Analyses of the two monkeys separately showed that the changes in slope and Shannon entropy were bigger in one of the two monkeys but went in the same direction (Supplementary File 2)”.

Anodal tDCS was only applied to anesthetized monkeys, which limits the conclusion that the authors are aiming for. It raises questions about the conclusion regarding brain state dependency. To address this, it would be better to include the cathodal tDCS session for anesthetized monkeys. If cathodal tDCS changes the connectivity during anesthesia, it becomes difficult to argue that the effects of cathodal tDCS vary depending on the state of consciousness as discussed in this paper. On the other hand, if cathodal tDCS would not produce any changes, the conclusion would then focus on the relationship between the polarity of tDCS and consciousness. In that case, the authors could maintain their conclusion but might need to refine it to reflect this specific relationship more accurately.

We agree with the reviewer that it would have been interesting to investigate the effects of cathodal tDCS in anesthetized monkeys. However, due to the challenging nature of the experimental procedures under anesthesia, we had to limit the investigations to only one stimulation modality. We chose to deliver anodal stimulation because, from a translational point of view, we aimed to provide new information on the effects of tDCS under anesthesia as a model for disorders of consciousness. It also made much more sense to increase the cortical excitability of the prefrontal cortex in an attempt to wake up the sedated monkeys rather than doing the opposite.

Actions in the text: We have added a new sentence in the Results section:

“Due to the challenging nature of the experimental procedures under anesthesia, we limited the investigations to only one stimulation modality. We chose to deliver anodal stimulation to provide new information on the effects of tDCS under anesthesia as a model for disorders of consciousness and to increase the cortical excitability of the PFC in an attempt to wake up the sedated monkeys.”

**Reviewer #3 (Public review):**
Summary:This study used transcranial direct current stimulation administered using small 'high-definition' electrodes to modulate neural activity within the non-human primate prefrontal cortex during both wakefulness and anaesthesia. Functional magnetic resonance imaging (fMRI) was used to assess the neuromodulatory effects of stimulation. The authors report on the modification of brain dynamics during and following anodal and cathodal stimulation during wakefulness and following anodal stimulation at two intensities (1 mA, 2 mA) during anaesthesia. This study provides some possible support that prefrontal direct current stimulation can alter neural activity patterns across wakefulness and sedation in monkeys. However, the reported findings need to be considered carefully against several important methodological limitations.Strengths:A key strength of this work is the use of fMRI-based methods to track changes in brain activity with good spatial precision. Another strength is the exploration of stimulation effects across wakefulness and sedation, which has the potential to provide novel information on the impact of electrical stimulation across states of consciousness.

We thank the Reviewer for the summary and for highlighting the strengths of our study.

Weaknesses:The lack of a sham stimulation condition is a significant limitation, for instance, how can the authors be sure that results were not affected by drowsiness or fatigue as a result of the experimental procedure?

We agree with the reviewer that adding control conditions could have strengthened our study. Control conditions usually consist of a sham condition or active control conditions. However, as mentioned in response to one of Reviewer 2 comments (R.2.5), we had to make choices as we could not perform as many experiments due to their demanding nature, especially under anesthesia.

In the awake state, we acquired data with two experimental conditions; the monkeys were exposed to either anodal (F4/O1) or cathodal (O1/F4) PFC tDCS. As anodal tDCS of the PFC induced only minor changes in brain dynamics, it could be considered as an active control condition for the cathodal condition, which had striking effects on the cortical dynamics. It is also worth noting that doubts have been raised about the neurobiological inertia of certain sham protocols. Indeed, different sham protocols have been employed in the literature, some of which may produce unintended effects (Fonteneau et al. 2019). Therefore, active control conditions, such as reversing the polarity of the stimulation or targeting a different brain region, have been proposed to provide better control (Fonteneau et al. 2019). Furthermore, in the context of experiments performed under anesthesia, the relevance of a sham control condition typically used to achieve adequate blinding is questionable.

With regard to drowsiness and fatigue as a result of the experimental procedure, we agree with the reviewer that this is a common problem in functional imaging due to the length of the recording sessions. We assumed, as was done in previous work (Uhrig, Dehaene, and Jarraya 2014; Wang et al. 2015), that the monkeys' performance on the fixation task during acquisition would capture these periods of fatigue. Therefore, only sessions with fixation rates above 85% were included in our analysis.

Actions in the text: We have now specified, in the Materials and Methods section, the fact that only runs with a high fixation rate (> 85%) were included in the study:

“To ensure that the results were not biased by fatigue or drowsiness due to the lengthy

In the anaesthesia condition, the authors investigated the effects of two intensities of stimulation (1 mA and 2 mA). However, a potential confound here relates to the possibility that the initial 1 mA stimulation block might have caused plasticity-related changes in neural activity that could have interfered with the following 2 mA block due to the lack of a sufficient wash-out period. Hence, I am not sure any findings from the 2 mA block can really be interpreted as completely separate from the initial 1 mA stimulation period, given that they were administered consecutively. Several previous studies have shown that same-day repeated tDCS stimulation blocks can influence the effects of neuromodulation (e.g., Bastani and Jaberzadeh, 2014, Clin Neurophysiol; Monte-Silva et al., J. Neurophysiology).

We agree with the reviewer’s comment that the initial 1 mA stimulation block might have induced changes in neural activity and that the 20-minute post 1 mA block would not be long enough to wash out these changes. This comment is very similar to the second comment made by Reviewer 2 (R.2.2). Although our experimental data do not support this possibility (as the differences between the 1 mA post-stimulation and baseline conditions were not significant), it is still conceivable that the stimulation produced some effects below the threshold of significance and that these might weakly contribute to the changes observed during and after the 2 mA stimulation.

Actions in the text: We have modified the paper according to the reviewers' comments (please see our answer and actions in the text to R.2.2.).

The different electrode placement for the two anaesthetised monkeys (i.e., Monkey R: F3/O2 montage, Monkey N: F4/O1 montage) is problematic, as it is likely to have resulted in stimulation over different brain regions. The authors state that "Because of the small size of the monkey's head, we expected that tDCS stimulation with these two symmetrical montages would result in nearly equivalent electric fields across the monkey's head and produce roughly similar effects on brain activity"; however, I am not totally convinced of this, and it really would need E-field models to confirm. It is also more likely that there would in fact be notable differences in the brain regions stimulated as the authors used HD-tDCS electrodes, which are generally more focal.

We thank the Reviewer for the remark, which is very similar to the second comment from Reviewer 2. Please see our answer to the first comment of Reviewer 2

Actions in the text: We have modified the paper according to the reviewers' comments (please see the actions taken in response to R.2.1.).

Given the very small sample size, I think it is also important to consider the possibility that some results might also be impacted by individual differences in response to stimulation. For instance, in the discussion (page 9, paragraph 2) the authors contrast findings observed in awake animals versus anaesthetised animals. However, different monkeys were examined for these two conditions, and there were only two monkeys in each group (monkeys J and Y for awake experiments [both male], and monkeys R and N [male and female] for the anaesthesia condition). From the human literature, it is well known that there is a considerable amount of inter-individual variability in response to stimulation (e.g., Lopez-Alonso et al., 2014, Brain Stimulation; Chew et al., 2015, Brain Stimulation), therefore I wonder if some of these differences could also possibly result from differences in responsiveness to stimulation between the different monkeys? At the end of the paragraph, the authors also state "Our findings also support the use of tDCS to promote rapid recovery from general anesthesia in humans...and suggest that a single anodal prefrontal stimulation at the end of the anesthesia protocol may be effective." However, I'm not sure if this statement is really backed-up by the results, which failed to report "any behavioural signs of awakening in the animals" (page 7)?

We thank the Reviewer for this comment. Because working with non-human primates is expensive and labor intensive, the sample sizes in classical macaque experiments are generally small (typically 2-4 subjects per experiment). Our sample size (i.e. 2 rhesus macaques in awake experiments and 2 macaques under sedation, 11 +/- 9 scan sessions per animal, 288 and 136 runs in the awake and anesthesia state, respectively) is comparable to other previous work in non-human primates using fMRI (Milham et al. 2018; Yacoub et al. 2020; Uchimura, Kumano, and Kitazawa 2024). In addition, we would like to point out that the baseline cortical dynamics we found before stimulation, whether in the awake or sedated state, are comparable to previous studies (Barttfeld et al. 2015; Uhrig et al. 2018; Tasserie et al. 2022). This suggests our results are reproducible across datasets, despite the small sample size.

That being said, we agree with the reviewer that inter-individual variability in response to stimulation can be considerable, as shown by a large body of literature in the field. It seems possible that the two monkeys studied in each condition responded differently to the stimulation. But even if that’s the case, our results suggest that at least in one of the two monkeys, cathodal PFC stimulation in the awake state and anodal PFC stimulation under propofol anesthesia induced striking changes in brain dynamics, which we believe is a significant contribution to the field.

In fact, supplementary analysis, as proposed by Reviewer 2 (cf R2.4), investigating how the different measurables we’ve used were differently affected by tDCS show that indeed monkey Y’s case is more apparent and significant than monkey J’s. Still, the effects observed in monkey J’s case are still congruent with what is observed in monkey Y’s and at the population level (though less flagrant). We also show that these inter-individual variabilities are outmatched by the inter-condition variability, (as indicated by our initially strong statistical results at the population levels), thus showing that, even though we have different responses depending on the subject, the effects observed at the population level cannot be only accounted for by the differences in subjects’ specificities.

Lastly, the Reviewer questioned whether our results support that a single anodal prefrontal stimulation at the end of the anesthesia protocol could effectively promote rapid recovery from general anesthesia, because the stimulation did not wake the animals in our experiments. It should be emphasized that in our case, the monkeys were stimulated while they were still receiving continuous propofol perfusion. In contrast, during the recovery process from anesthesia, the delivery of the anesthetic drug is stopped. It is therefore conceivable that anodal PFC tDCS, which successfully enriched brain dynamics in sedated monkeys in our experiments, may accelerate the recovery from anesthesia when the drug is no longer administered.

Actions in the text: We have added a line in the Materials and Methods to compare to other studies:

“Our sample size is comparable to previous work in NHP using fMRI (Milham et al. 2018; Yacoub et al. 2020; Uchimura, Kumano, and Kitazawa 2024).”

**Reviewing Editor Comments:**
In some cases, authors opt to submit a revised manuscript. Should you choose to do so, please be aware that the reviewers have indicated that their appraisal is unlikely to change unless some of the suggested field modelling is incorporated into the work. This may change the evaluation of the strength of evidence, but the final wording will be subject to reviewer discretion. Details for responding to the reviews are provided at the bottom of this email.
**Reviewer #1 (Recommendations for the authors):**
The work should discuss the implications of their experiments for using tDCS to arouse a patient from a coma. The anesthetized animal is effectively in a drug-induced coma. While they observed connectivity changes, these changes did not map nicely onto behavioral changes.

I would suggest that the authors spell out more clearly what they view as the clinical implications of their work in terms of new insights into how tDCS may be used to either understand and or treat disorders of consciousness.

We thank the Reviewer for his thoughtful comments. We appreciate the opportunity to clarify and expand on the key findings and implications of our work, particularly regarding the new insights into how tDCS can be used to understand and treat disorders of consciousness. We therefore provide a broader perspective on the clinical implications of our experiments regarding coma and disorders of consciousness. We also agree with the Reviewer that the absence of behavioral changes but the presence of functional differences should be more clearly addressed.

Actions in the text: We have added a few lines about the relevance of anesthesia as a model for disorders of consciousness in the Introduction part:

“Anesthesia provides a unique model for studying consciousness, which, similarly to DOC, is characterized by the disruption or even the loss of consciousness (Luppi 2024). Additionally, anesthesia mechanisms involve several subcortical nuclei that are key components of the brain's sleep and arousal circuits (Kelz and Mashour 2019).”

In the Discussion section, we have modified and expanded a paragraph about the effects of tDCS in DOC patients and how this technique could be further used to study consciousness: From another clinical perspective, our results demonstrating that 2 mA anodal PFC tDCS decreased the structure-function correlation and modified the dynamic repertoire of brain patterns during anesthesia (Figures 6 and 7) are consistent with the beneficial effects of such stimulation in DOC patients (Thibaut et al., 2014; Angelakis et al., 2014; Thibaut et al., 2017; Zhang et al., 2017; Martens et al., 2018; Cavinato et al., 2019; Wu et al., 2019; Hermann et al., 2020; Peng et al., 2022; Thibaut et al., 2023). Although some clinical trials investigated the effects of stimulating other brain regions, such as the motor cortex (Martens et al., 2019; Straudi et al., 2019) or the parietal cortex (Huang et al., 2017; Guo et al., 2019; Zhang et al., 2022; Wan et al., 2023; Wang et al., 2020), the DLPFC appears to be the most effective target for patients with a minimally conscious state (Liu et al., 2023). In terms of neuromodulatory effects in DOC patients, DLPFC tDCS has been reported to increase global excitability (Bai et al., 2017), increase the P300 amplitude (Zhang et al., 2017; Hermann et al., 2020), improve the fronto-parietal coherence in the theta band (Bai et al., 2018), enhance the putative EEG markers of consciousness (Bai et al., 2018; Hermann et al., 2020) and reduce the incidence of slow-waves in the resting state (Mensen et al., 2020). Our findings further support the PFC as a relevant target for modulating consciousness level and align with growing evidence showing that the PFC plays a key role in conscious access networks (Mashour, Pal, and Brown 2022; Panagiotaropoulos 2024). Nevertheless, we hypothesize that other brain targets for tDCS may be of interest for consciousness restoration, potentially using multi-channel tDCS (Havlík et al., 2023). Among transcranial electrical stimulation techniques, tDCS has the great advantage of facilitating either excitation or inhibition of brain regions, depending on the polarity of the stimulation (Sdoia et al., 2019) exploited this advantage to investigate the causal involvement of the DLPFC in conscious access to a visual stimulus during an attentional blink paradigm. While conscious access was enhanced by anodal stimulation of the left DLPFC compared to sham stimulation, opposite effects were found with cathodal stimulation compared to sham over the same locus. Finally, this literature and our findings suggest that tDCS constitutes a non-invasive, reversible, and powerful tool for studying consciousness.”

We have added a new paragraph about patients with cognitive-motor dissociation and dissociation between consciousness and behavioral responsiveness:

“Changes in the state of consciousness are generally closely associated with changes in behavioural responsiveness, although some rare cases of dissociation have been described. Cognitive-motor dissociation (CMD) is a condition observed in patients with severe brain injury, characterized by behavior consistent with unresponsive wakefulness syndrome or a minimally conscious state minus (Thibaut et al., 2019). However, in these patients, specific cortical brain areas activate in response to mental imagery tasks (e.g., imagining playing tennis or returning home) in a manner indistinguishable from that of healthy controls, as shown through fMRI or EEG (Thibaut et al., 2019; Owen et al., 2006; Monti et al., 2010; Bodien et al., 2024). Thus, although CMD patients are behaviorally unresponsive, they demonstrate cognitive awareness that is not outwardly apparent. It is worth noting that both the structure-function correlation and the rate of the pattern closest to the anatomy were shown to be significantly reduced in unresponsive patients showing command following during mental imagery tasks compared to those who do not show command following (Demertzi et al., 2019). These observations would be compatible with our findings in anesthetized macaques exposed to 2 mA anodal PFC tDCS. The richness of the brain dynamics would be recovered (at least partially, in our experiments), but not the behaviour. This hypothesis also fits with a recent longitudinal fMRI study on patients recovering from coma (Crone et al., 2020). The researchers examined two groups of patients: one group consisted of individuals who were unconscious at the acute scanning session but regained consciousness and improved behavioral responsiveness a few months later, and the second group consisted of patients who were already conscious from the start and only improved behavioral responsiveness at follow-up. By comparing these two groups, the authors could distinguish between the recovery of consciousness and the recovery of behavioral responsiveness. They demonstrated that only initially conscious patients exhibited rich brain dynamics at baseline. In contrast, patients who were unconscious in the acute phase and later regained consciousness had poor baseline dynamics, which became more complex at follow-up. Complete recovery of both consciousness and responsiveness under general anesthesia is possible through electrical stimulation of the central thalamus (Redinbaugh et al., 2020; Tasserie et al., 2022).”

**Reviewer #2 (Recommendations for the authors):**
Method(1) The authors mentioned that they used HD-tDCS in their experiments; however, they used 1 x 1 tDCS, which is not HD-tDCS but rather single-channel tDCS.

We thank the Reviewing Editor for pointing out this ambiguous wording. We understand that "HD-tDCS", which we used in our paper to refer to high-density 1x1 tDCS (because we used small carbon electrodes instead of the large sponge electrodes employed in conventional tDCS), may cause some confusion with high-definition tDCS, which uses compact ring electrodes and most commonly refers to a 4x1 montage (1 active central electrode over the target area and 4 return electrodes placed around the central electrode).

Therefore, to avoid any confusion, we will use the term "tDCS" rather than “HD-tDCS” to qualify the technique used in this paper and suppress mentions of high-density or high-definition tDCS.

Actions in the text: We have replaced the abbreviation “HD-tDCS” with “tDCS” throughout the paper. We have also suppressed the sentence about high-definition tDCS in the Introduction (“While conventional tDCS relies on the use of relatively large rectangular pad electrodes, high-density tDCS (HD-tDCS) utilizes more compact ring electrodes, allowing for increased focality, stronger electric fields, and presumably, greater neurophysiological changes (Datta et al. 2009; Dmochowski et al. 2011)”) and the two related citations in the References section.

(2) Please provide the characteristics of electrodes, including their size, shape, and thickness.

We thank the Reviewing Editor for this recommendation. We now provide the complete characteristics of the tDCS electrodes used in the paper.

Actions in the text: We have added a sentence describing the characteristics of the tDCS electrodes in the Materials and Methods section:

“We used a 1x1 electrode montage with two carbon rubber electrodes (dimensions: 1.4 cm x 1.85 cm, 0.93 cm thick) inserted into Soterix HD-tES MRI electrode holders (base diameter: 25 mm; height: 10.5 mm), which are in contact with the scalp. These electrodes (2.59 cm2) are smaller than conventional tDCS sponge electrodes (typically 25 to 35 cm^2^).”

(3) Could the authors clarify why they chose to stimulate the right DLPFC? Is there a specific rationale for this choice? Additionally, could the authors explain how they ensured that the stimulation targeted the DLPFC, given that the monkey cap might differ from human configurations? In many NHP studies, structural MRI is used to accurately determine electrode placement. Considering that a single channel F4 - O2 montage was used, even a small displacement of the frontal electrode laterally could result in the electric field not adequately covering the DLPFC. Could the authors provide structural MRI images and details of electrode positioning to help readers better understand targeting accuracy?

We thank the Reviewing Editor for the thoughtful comments and recommendations. We appreciate the opportunity to further clarify our rationale for stimulating the right DLPFC and also the suggestion to provide structural MRI images and details of electrode positioning, which we think will improve the quality of the paper by showing targeting accuracy.

First, we would like to clarify that our initial decision to stimulate the right PFC in most animals was driven by experimental constraints. Indeed, we had limited access to the left PFC in three of the four macaques, either due to the presence of cement (spreading asymmetrically from the centre of the head) used to fix the head post in awake animals or due to a scar in one of the two animals studied under anesthesia.

Second, we agree with the Reviewing Editor on the importance of showing details of electrode positioning and evidence of targeting accuracy across MRI sessions. Therefore, we now provide structural images showing the positions of anodal and cathodal electrodes in almost all acquired sessions: 10 sessions (out of 10) under anesthesia and 30 sessions in the awake state (out of 34 sessions, because we could not acquire structural images in four sessions). These images show that, in anesthesia experiments, the anodal electrode was positioned over the dorsal prefrontal cortex and the cathodal electrode was placed over the contralateral occipital cortex (at the level of the parieto–occipital junction) in both monkeys. In the awake state, the montage still targeted the prefrontal cortex and the occipital cortex, but with a slightly different placement. One of the electrodes was placed over the prefrontal cortex, closer to the premotor cortex than in anesthesia experiments, while the other one was placed over the occipital cortex (V1), slightly more posterior than in anesthesia experiments. These images therefore show that the placement was relatively accurate across sessions and reproducible between monkeys in each of the two arousal conditions.

Actions in the text: We have added a supplementary file showing electrode positioning in 40 of the 44 acquired MRI sessions (Supplementary File 1). We have also added a new supplement figure (Figure 1 - figure supplement 1) showing electrode positioning in representative MRI sessions of the awake and anesthetized experiments in the main manuscript.

We added a few sentences referring to these figures in the Result section:

“Representative structural images showing electrode placements on the head of the two awake monkeys are shown in Figure 1 - figure supplement 1A. Supplementary File 1 displays the complete set of structural images, showing that the two electrodes were accurately placed over the prefrontal cortex and the occipital cortex in a reproducible manner across awake sessions.”

Figure 1 - figure supplement 1. Structural images displaying electrode placements on the head of monkeys. (A) Awake experiments. Representative sagittal, coronal and transverse MRI sections, and the corresponding skin reconstruction images showing the position of the prefrontal and the occipital electrodes on the head of monkeys J. and Y. (B) Anesthesia experiments. Representative sagittal, coronal and transverse MRI sections, and the corresponding skin reconstruction images showing the position of the prefrontal and occipital electrodes over the occipital cortex on the head of monkeys R. and N.

**S**upplementary File 1 (see attached file). Structural images showing the position of the tDCS electrodes on the monkey's head across sessions. Sagittal, coronal and transverse MRI sections, and corresponding skin reconstruction images showing the position of the prefrontal and occipital electrodes on the monkey's head for each MRI session (except for 4 sessions in which no anatomical scan was acquired). The two electrodes were accurately placed over the prefrontal cortex and the occipital cortex in a reproducible manner across sessions and between the two monkeys studied in each arousal state. In anesthesia experiments, the anodal electrode was placed over the dorsal prefrontal cortex, while the cathodal electrode was positioned over the parieto-occipital junction. In awake experiments, the prefrontal electrode was positioned over the dorsal prefrontal cortex/pre-motor cortex, while the occipital electrode was placed over the visual area 1. The position of the two electrodes differed slightly between the anesthetized and awake experiments due to different body positions (the prone position of the sedated monkeys prevented a more posterior position of the occipital electrode) and also due to the presence of a headpost on the head of the two monkeys in awake experiments (the monkeys we worked with in anesthesia experiments did not have an headpost).

(4) If the authors did not analyze the data for the passive event-related auditory response, it may be helpful to remove the related sentence to avoid potential confusion for readers.

We thank the Reviewing Editor for the comment. Although we understand the reviewer’s point of view, we decide to keep this information in the paper to inform the reader that the macaques were passively engaged in an auditory task, as this could have some influence on the brain state. In the Materials and Methods section, we already mentioned that the analysis of the cerebral responses to the auditory paradigm is not part of the paper. We have modified the sentence to make it clearer and to avoid potential confusion for readers.

Actions in the text: We have modified the sentence referring to the passive event-related auditory response in the Materials and Methods section:

“All fMRI data were acquired while the monkeys were engaged in a passive event-related auditory task, the local-global paradigm, which is based on local and global deviations from temporal regularities (Bekinschtein et al. 2009; Uhrig, Dehaene, and Jarraya 2014). The present paper does not address how tDCS perturbs cerebral responses to local and global deviants, which will be the subject of future work.”

(5) Could the authors clarify what x(t) represents in the equation? Additionally, it would be better to number the equations.

We apologize for the confusion, x(t) represents the evolution of the BOLD signals over time. We have numbered the equations as suggested.

Actions in the text: We have added explanations about the notation and numerotation of equations.

(6) It would be much better to provide schematic illustrations to explain what the authors did for analyzing fMRI data.

We thank the Reviewing Editor for the suggestion and now provide a new figure as suggested.

Actions in the text: We have added a new figure (Figure 2) graphically showing the overall analysis performed. We have added a sentence about the new Figure 2 in the Results section: “A graphical overview of the overall analysis is shown in Figure 2.” We have renumbered Figure 2 - supplement figures accordingly.

Figure 2. fMRI Phase Coherence analysis. (A) (Left) Animals were scanned before, during and after PFC tDCS stimulation in the awake state (two macaques) or under deep propofol anesthesia (two macaques). Right Example of Z-scored filtered BOLD time series for one macaque, 111 time points with a TR of 2.4 s. (B) Hilbert transform of the z-scored BOLD signal of one ROI into its time-varying amplitude A(t) (red) and the real part of the phase φ (green). In blue, we recover the original z-scored BOLD signal as A(t)cos(φ). (C) Example of the phase of the Hilbert transform for each brain region at one TR. (D) Symmetric matrix of cosines of the phase differences between all pairs of brain regions. (E) We concatenated the vectorized form of the triangular superior of the phase difference matrices for all TRs for all participants, in all the conditions for both datasets separately obtaining using the K-means algorithm, the brain patterns whose statistics are then analyzed in the different conditions.

Results(1) In Figures 3A, 5A, and 6A showing brain connectivity, it is difficult to relate the connectivity variability among the brain regions. Instead of displaying connection lines for nodes, it would be more effective if the authors highlighted significant, strong connectivity within specific brain regions using additional methods, such as bootstrapping.

We thank the Reviewing Editor for the comment and suggestion. The connection lines indeed represent all the synchronizations above 0.5 and all the anti-synchronization below -0.5 between all pairs of brain regions. As suggested, another element we haven’t addressed is the heterogeneity in coherences between individual brain regions. We hence propose additional supplementary figures showing, for all centroids mentioned in main figures, the variance in phase-based connectivity of the distributions of coherence of all brain regions to the rest of the brain. High value would then indicate a wide range of values of coherence, while low would indicate the different coherence a region has with the rest of the brain have similar values. Thus, a brain with uniform color would indicate high homogeneity in coherence among brain regions, while sharp changes in colors would reveal that certain regions are more subject to high variance in their coherence distributions. We expect this new figure to more clearly expose the connectivity variability among the brain regions.

Actions in the text: We have added new figures showing, for all centroids mentioned in the main figures, the variances in phase-based connectivity of the distributions of coherence (Figure 3 - figure supplement 3; Figure 5 - figure supplement 2; Figure 6 - figure supplement 3; Figure 7 - figure supplement 2). One of them is shown below for the only awake analysis (Figure 3 - figure supplement 3).

Figure 3 - figure supplement 3. Variance in inter-region phase coherences of brain patterns. Low values (red and light red) indicate that the distribution of synchronizations between a brain region and the rest of the brain has relatively low variance, while high values (blue and light blue) indicate relatively high variance. Are displayed both supra (top) and subdorsal (bottom) views for each brain pattern from the main figure, ordered similarly as previously: from left (1) to right (6) as their respective SFC increases.

We added a few sentences about variances in phase-based connectivity of the distributions of coherence in the Result section:

“Further investigation of the variances in inter-region phase coherences of brain patterns, presented in Figure 3 - figure supplement 3, revealed two main findings. First, all the patterns exhibited some degree of lateral symmetry. Second, except for the pattern with the highest SFC, most patterns displayed high heterogeneity in their coherence variances and striking inter-pattern differences. These observations reflect both the segmentation of distinct functional networks across patterns and a topological organization within the patterns themselves: some regions showed a broader spectrum of synchrony with the rest of the brain, while others exhibited narrower distributions of coherence variances. For instance, unlike other brain patterns, pattern 5 was characterized by a high coherence variance in the frontal premotor areas and low variance in the occipital cortex, whereas pattern 3 had a high variance in the frontal and orbitofrontal regions. In addition, we performed the main analyses separately for the two monkeys, explored the inter-condition variability (Supplementary File 2), and computed classical measures of functional connectivity such as average FC matrices and functional graph properties (modularity, efficiency and density) of the visited FC states (Supplementary File 3).”

“The variance in inter-regional phase coherence across brain patterns showed notably that pattern 4, in contrast to most other patterns, was characterized by a high variance in frontal premotor areas and a low variance in the occipital cortex (Figure 5 - figure supplement 2)."

“The variance in inter-region phase coherences of the brain patterns is displayed in Figure 6 - figure supplement 3 and showed a striking heterogeneity between the patterns. For example, pattern 5 had a low overall variance (except in the frontal cortex), while pattern 1 was the only pattern with a high variance in the occipital cortex.”

“The variance in inter-region phase coherences of brain patterns is displayed in Figure 6 - figure supplement 2.”

(2) For both conditions, only 2 to 3 out of 6 patterns showed significant effects of tDCS on the occurrence rate. Is it sufficient to claim the authors' conclusion?

We thank the Reviewer Editor for the comment. We would like to point out that similar kinds of differences in the occurrence rates of specific brain patterns (particularly in patterns at the extremities of the SFC scale) have already been reported previously. Prior works in patients suffering from disorders of consciousness, in healthy humans or in non-human primates, have shown, by using a similar method of analysis, that not all brain states are equally disturbed by loss of consciousness, even in different modalities of unconscious transitioning (Luppi et al. 2021; Z. Huang et al. 2020; Demertzi et al. 2019; Castro et al. 2023; Golkowski et al. 2019; Barttfeld et al. 2015). Therefore, yes we believe that our conclusions are still supported by the results.

(3) If the authors want to assert that the brain state significantly influences the effects of tDCS as discussed in the manuscript, further analysis is necessary. First, it would be great to show the difference in connectivity between two consciousness conditions during the baseline (resting state) to see how resting state connectivity or structural connectivity varies. Second, demonstrating the difference in connectivity between the awake and anesthetized conditions (e.g., awake during cathodal vs. anesthetized cathodal) to show how the connectivity among the brain regions was changed by the brain state during tDCS. This would strengthen the authors' conclusion.

We thank the reviewer for this comment. Firstly, we’d like to clarify that the structural connectivity doesn’t change from one session to another in the same animal and minimally between subjects. Secondly, we agree with the Reviewing Editor that it is informative to show the differences between the baselines and this is what we have done. The results are shown in Figures 5 and 7. Regarding the comparison of the stimulating conditions across arousal levels, the only contrast that we could make is to compare 2 mA anodal awake with 2 mA anodal anesthetized (during and post-stimulation). However, as 2 mA anodal stimulation in the awake state did not affect the connectivity much (compared to the awake baseline), the results would be almost similar to the comparison of the awake baseline with 2 mA anodal anesthetized, which is shown in Figure 7. Therefore, we believe that this would result in minimal informative gains and even more redundancy.

**Reviewer #3 (Recommendations for the authors):**
Introduction, par 2: HD-tDCS does not necessarily produce stronger electric fields (E-fields) in the brain. The E-field is largely montage-dependent, and some configurations such as the 4x1 configuration can actually have weaker E-fields compared to conventional tDCS designs (i.e., with two sponge electrodes) as electrodes are often closer together resulting in more current being shunted by skull, scalp, and CSF. I would consider re-phrasing this section.

We agree with the Reviewer Editor that high-definition tDCS does not necessarily produce stronger electric fields in the brain and apologize for the confusion caused by our use of HD-tDCS to refer to high-density tDCS. To avoid any confusion, we have removed the sentence mentioning that HD-tDCS produces stronger electric fields.

Actions in the text: We have removed the sentence about high-definition tDCS in the Introduction (“While conventional tDCS relies on the use of relatively large rectangular pad electrodes, high-density tDCS (HD-tDCS) utilizes more compact ring electrodes, allowing for increased focality, stronger electric fields, and presumably, greater neurophysiological changes (Datta et al. 2009; Dmochowski et al. 2011)”) and the two related citations in the References section.